# 3D GENERATION ON IMAGENET

**Ivan Skorokhodov**[*]     **Aliaksandr Siarohin**     **Yinghao Xu**[*]     **Jian Ren**     **Hsin-Ying Lee**
KAUST                        Snap Inc.                   CUHK                  Snap Inc.        Snap Inc.

**Peter Wonka**                                        **Sergey Tulyakov**
KAUST                                                  Snap Inc.

## ABSTRACT

All existing 3D-from-2D generators are designed for well-curated single-category datasets, where all the objects have (approximately) the same scale, 3D location and orientation, and the camera always points to the center of the scene. This makes them inapplicable to diverse, in-the-wild datasets of non-alignable scenes rendered from arbitrary camera poses. In this work, we develop *3D generator with Generic Priors (3DGP)*: a 3D synthesis framework with more general assumptions about the training data, and show that it scales to very challenging datasets, like ImageNet. Our model is based on three new ideas. First, we incorporate an *inaccurate* off-the-shelf depth estimator into 3D GAN training via a special depth adaptation module to handle the imprecision. Then, we create a flexible camera model and a regularization strategy for it to learn its distribution parameters during training. Finally, we extend the recent ideas of transferring knowledge from pretrained classifiers into GANs for patch-wise trained models by employing a simple distillation-based technique on top of the discriminator. It achieves more stable training than the existing methods and speeds up the convergence by at least 40%. We explore our model on four datasets: SDIP Dogs $256^2$, SDIP Elephants $256^2$, LSUN Horses $256^2$, and ImageNet $256^2$ and demonstrate that 3DGP outperforms the recent state-of-the-art in terms of both texture and geometry quality.

Code and visualizations: https://snap-research.github.io/3dgp

## 1 INTRODUCTION

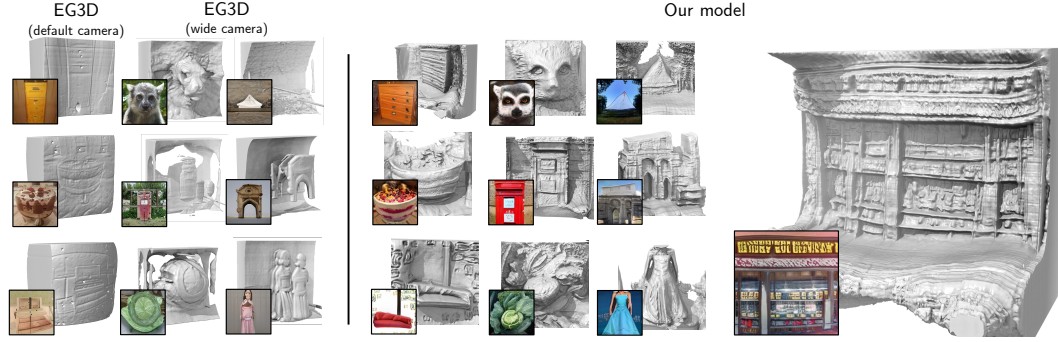

Figure 1: Selected samples from EG3D (Chan et al., 2022) and our generator trained on ImageNet $256^2$ (Deng et al., 2009). EG3D models the geometry in low resolution and renders either flat shapes (when trained with the default camera distribution) or repetitive "layered" ones (when trained with a wide camera distribution). In contrast, our model synthesizes the radiance field in the full dataset resolution and learns high-fidelity details during training. Zoom-in for a better view.

We witness remarkable progress in the domain of 3D-aware image synthesis. The community is developing new methods to improve the image quality, 3D consistency and efficiency of the generators

---

[*]Work done during internship at Snap Inc.

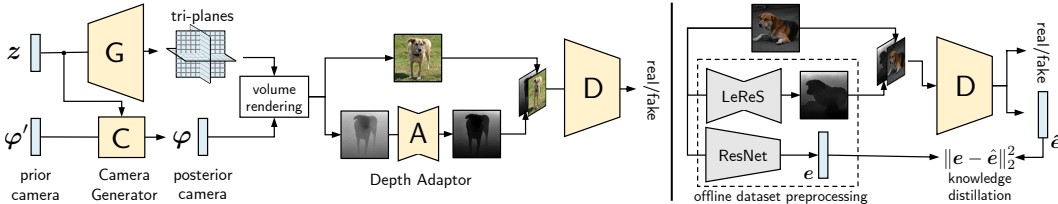

Figure 2: **Model overview**. Left: our tri-plane-based generator. To synthesize an image, we first sample camera parameters from a prior distribution and pass them to the camera generator. This gives the posterior camera parameters, used to render an image and its depth map. The depth adaptor mitigates the distribution gap between the rendered and the predicted depth. Right: our discriminator receives a 4-channel color-depth pair as an input. A fake sample consists of the RGB image and its (adapted) depth map. A real sample consists of a real image and its estimated depth. Our two-headed discriminator predicts adversarial scores and image features for knowledge distillation.

(e.g., Chan et al. (2022); Deng et al. (2022); Skorokhodov et al. (2022); Zhao et al. (2022); Schwarz et al. (2022)). However, all the existing frameworks are designed for well-curated and aligned datasets consisting of objects of the same category, scale and global scene structure, like human or cat faces (Chan et al., 2021). Such curation requires domain-specific 3D knowledge about the object category at hand, since one needs to infer the underlying 3D keypoints to properly crop, rotate and scale the images (Deng et al., 2022; Chan et al., 2022). This makes it infeasible to perform a similar alignment procedure for large-scale multi-category datasets that are inherently "non-alignable": there does not exist a single canonical position which all the objects could be transformed into (e.g., it is impossible to align a landscape panorama with a spoon).

To extend 3D synthesis to in-the-wild datasets, one needs a framework which relies on more universal 3D priors. In this work, we make a step towards this direction and develop a 3D generator with Generic Priors (3DGP): a 3D synthesis model which is guided only by (imperfect) depth predictions from an off-the-shelf monocular depth estimator. Surprisingly, such 3D cues are enough to learn reasonable scenes from loosely curated, non-aligned datasets, such as ImageNet (Deng et al., 2009).

Training a 3D generator on in-the-wild datasets comes with three main challenges: 1) extrinsic camera parameters of real images are unknown and impossible to infer; 2) objects appear in different shapes, positions, rotations and scales, complicating the learning of the underlying geometry; and 3) the dataset typically contains a lot of variation in terms of texture and structure, and is difficult to fit even for 2D generators. As shown in Fig 1 (left), state-of-the-art 3D-aware generators, such as EG3D (Chan et al., 2022), struggle to learn the proper geometry in such a challenging scenario. In this work, we develop three novel techniques to address those problems.

*Learnable "Ball-in-Sphere" camera distribution*. Most existing methods utilize a restricted camera model (e.g., (Schwarz et al., 2020; Niemeyer & Geiger, 2021b; Chan et al., 2021)): the camera is positioned on a sphere with a constant radius, always points to the world center and has fixed intrinsics. But diverse, non-aligned datasets violate these assumptions: e.g., dogs datasets have images of both close-up photos of a snout and photos of full-body dogs, which implies the variability in the focal length and look-at positions. Thus, we introduce a novel camera model with 6 degrees of freedom to address this variability. We optimize its distribution parameters during training and develop an efficient gradient penalty for it to prevent its collapse to a delta distribution.

*Adversarial depth supervision (ADS)*. A generic image dataset features a wide diversity of objects with different shapes and poses. That is why learning a meaningful 3D geometry together with the camera distribution is an ill-posed problem, as the incorrect scale can be well compensated by an incorrect camera model (Hartley & Zisserman, 2003), or flat geometry (Zhao et al., 2022; Chan et al., 2022). To instill the 3D bias, we provide the scene geometry information to the discriminator by concatenating the depth map of a scene as the 4-th channel of its RGB input. For real images, we use their (*imperfect*) estimates from a generic off-the-shelf monocular depth predictor (Miangoleh et al., 2021). For fake images, we render the depth from the synthesized radiance field, and process it with a shallow depth adaptor module, bridging the distribution gap between the estimated and rendered depth maps. This ultimately guides the generator to learn the proper 3D geometry.

*Knowledge distillation into Discriminator*. Prior works found it beneficial to transfer the knowledge from off-the-shelf 2D image encoders into a synthesis model (Sauer et al., 2022). They typically

utilize pre-trained image classifiers as the discriminator backbone with additional regularization strategies on top (Sauer et al., 2021; Kumari et al., 2022). Such techniques, however, are only applicable when the discriminator has an input distribution similar to what the encoder was trained on. This does not suit the setup of patch-wise training (Schwarz et al., 2020) or allows to provide depth maps via the 4-th channel to the discriminator. That is why we develop a more general and efficient knowledge transfer strategy based on knowledge distillation. It consists in forcing the discriminator to predict features of a pre-trained ResNet50 (He et al., 2016) model, effectively transferring the knowledge into our model. This technique has just 1% of computational overhead compared to standard training, but allows to improve FID for both 2D and 3D generators by at least 40%.

First, we explore our ideas on *non-aligned* single-category image datasets: SDIP Dogs $256^2$ (Mokady et al., 2022), SDIP Elephants $256^2$ (Mokady et al., 2022), and LSUN Horses $256^2$ (Yu et al., 2015). On these datasets, our generator achieves better image appearance (measured by FID Heusel et al. (2017)) and geometry quality than the modern state-of-the-art 3D-aware generators. Then, we train the model on all the $1,000$ classes of ImageNet (Deng et al., 2009), showing that multi-categorical 3D synthesis is possible for non-alignable data (see Fig. 1).

## 2 RELATED WORK

**3D-aware image synthesis**. Mildenhall et al. (2020) introduced Neural Radiance Fields (NeRF): a neural network-based representation of 3D volumes which is learnable from RGB supervision only. It ignited many 3D-aware image/video generators (Schwarz et al., 2020; Niemeyer & Geiger, 2021b; Chan et al., 2021; Xue et al., 2022; Zhou et al., 2021; Wang et al., 2022; Gu et al., 2022; Or-El et al., 2021; Chan et al., 2022; Skorokhodov et al., 2022; Zhang et al., 2022; Xu et al., 2021; Bahmani et al., 2022), all of them being GAN-based (Goodfellow et al., 2014). Many of them explore the techniques to reduce the cost of 3D-aware generation for high-resolution data, like patch-wise training (e.g., Schwarz et al. (2020); Meng et al. (2021); Skorokhodov et al. (2022)), MPI-based rendering (e.g., Zhao et al. (2022)) or training a separate 2D upsampler (e.g., Gu et al. (2022)).

**Learning the camera poses**. NeRFs require known camera poses, obtained from multi-view stereo (Schönberger et al., 2016) or structure from motion (Schönberger & Frahm, 2016). Alternatively, a group of works has been introduced to either automatically estimate the camera poses (Wang et al., 2021) or finetune them during training (Lin et al., 2021; Kuang et al., 2022). The problem we are tackling in this work is fundamentally different as it requires learning not the camera poses from multi-view observations, but a *distribution* of poses, while having access to sparse, single-view data of diverse object categories. In this respect, the work closest to ours is CAMPARI (Niemeyer & Geiger, 2021a) as it also learns a camera distribution.

**GANs with external knowledge.** Several works observed improved convergence and fidelity of GANs when using existing, generic image-based models (Kumari et al., 2022; Sauer et al., 2021; 2022; Mo et al., 2020), the most notable being StyleGAN-XL (Sauer et al., 2022), which uses a pre-trained EfficientNet (Tan & Le, 2019) followed by a couple of discriminator layers. A similar technique is not suitable in our case as pre-training a generic RGB-D network on a large-scale RGB-D dataset is problematic due to the lack of data. Another notable example is FreezeD Mo et al. (2020), which proposes to distill discriminator features for GAN finetuning. Our work, on the other hand, relies on an existing model for image classification.

**Off-the-shelf depth guidance**. GSN (DeVries et al., 2021) also concatenates depth maps as the 4-th channel of the discriminator's input, but they utilize ground truth depths, which are not available for large-scale datasets. DepthGAN (Shi et al., 2022) uses predictions from a depth estimator to guide the training of a 2D GAN. Exploiting monocular depth estimators for improving neural rendering was also explored in concurrent work (Yu et al., 2022), however, their goal is just geometry reconstruction. The core characteristic of our approach is taking the depth estimator imprecision into account by training a depth adaptor module to mitigate it (see §3.2).

## 3 METHOD

We build our generator on top of EpiGRAF (Skorokhodov et al., 2022) since its fast to train, achieves reasonable image quality and does not need a 2D upsampler, relying on patch-wise training in-

stead (Schwarz et al., 2020). Given a random latent code $z$, our generator G produces a tri-plane representation for the scene. Then, a shallow 2-layer MLP predicts RGB color and density $\sigma$ values from an interpolated feature vector at a 3D coordinate. Then, images and depths are volumetrically rendered (Mildenhall et al., 2020) at any given camera position. Differently to prior works (Chan et al., 2021; Niemeyer & Geiger, 2021b) that utilize fixed camera distribution, we sample camera from a trainable camera generator C (see §3.1). We render depth and process it via the depth adaptor (see §3.2), bridging the domains of rendered and estimated depth maps (Miangoleh et al., 2021). Our discriminator D follows the architecture of StyleGAN2 (Karras et al., 2020a), additionally taking either adapted or estimated depth as the 4-th channel. To further improve the image fidelity, we propose a simple knowledge distillation technique, that enriches D with external knowledge obtained from ResNet (He et al., 2016) (see §3.3). The overall model architecture is shown in Fig. 2.

### 3.1 LEARNABLE "BALL-IN-SPHERE" CAMERA DISTRIBUTION

**Limitations of Existing Camera Parameterization.** The camera parameterization of existing 3D generators follows an overly simplified distribution — its position is sampled on a fixed-radius sphere with fixed intrinsics, and the camera always points to $(0, 0, 0)$. This parametrization has only two degrees of freedom: pitch and yaw ($\varphi_{\text{pos}}$ in Fig. 3 (a)), implicitly assuming that all the objects could be centered, rotated and scaled with respect to some canonical alignment. However, natural in-the-wild 3D scenes are inherently non-alignable: they could consist of multiple objects, objects might have drastically different shapes and articulation, or they could even be represented only as volumes (like smoke). This makes the traditional camera conventions ill-posed for such data.

**Learnable "Ball-in-Sphere" Camera Distribution.** We introduce a new camera parametrization which we call "Ball-in-Sphere". Contrary to the standard one, it has four additional degrees of freedom: the field of view $\varphi_{\text{fov}}$, and pitch, yaw, and radius of the inner sphere, specifying the look-at point within the outer sphere ($\varphi_{\text{lookat}}$ in Fig. 3 (b)). Combining with the standard parameters on the outer sphere, our camera parametrization has six degrees of freedom $\varphi = [\varphi_{\text{pos}} \parallel \varphi_{\text{fov}} \parallel \varphi_{\text{lookat}}]$, where $\parallel$ denotes concatenation.

Instead of manually defining camera distributions, we learn the camera distribution during training for each dataset. In particular, we train the camera generator network C that takes camera parameters sampled from a sufficiently wide camera prior $\varphi'$ and produces new camera parameters $\varphi$. For a class conditional dataset, such as ImageNet where scenes have significantly different geometry, we additionally condition this network on the class label $c$ and the latent code $z$, i.e. $\varphi = \mathsf{C}(\varphi', z, c)$. For a single category dataset we use $\varphi = \mathsf{C}(\varphi', z)$.

**Camera Gradient Penalty.** To the best of our knowledge, CAMPARI (Niemeyer & Geiger, 2021a) is the only work which also learns the camera distribution. It samples a set of camera parameters from a wide distribution and passes it to a shallow neural network, which produces a residual $\Delta\varphi = \varphi - \varphi'$ for these parameters. However, we observed that such a regularization is too weak for the complex datasets we explore in this work, and leads to a distribution collapse (see Fig. 7). Note, that a similar observation was also made by Gu et al. (2022).

To prevent the camera generator from producing collapsed camera parameters, we seek a new regularization strategy. Ideally, it should prevent constant solutions, while at the same time, reducing the Lipschitz constant for C, which is shown to be crucial for stable training of generators (Odena et al., 2018). We can achieve both if we push the derivatives of the predicted camera parameters with respect to the prior camera parameters to either one or minus one,

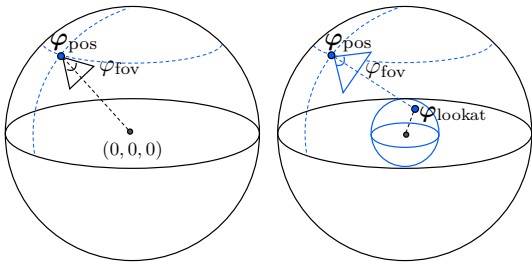

(a) Standard camera model (b) Our camera model

Figure 3: **Camera model**. (a) Conventional camera model is designed for aligned datasets and uses just 2 degrees of freedom. (b) The proposed "Ball-in-Sphere" parametrization has 4 additional degrees of freedom: field of view and the look at position. Variable parameters are shown in blue.

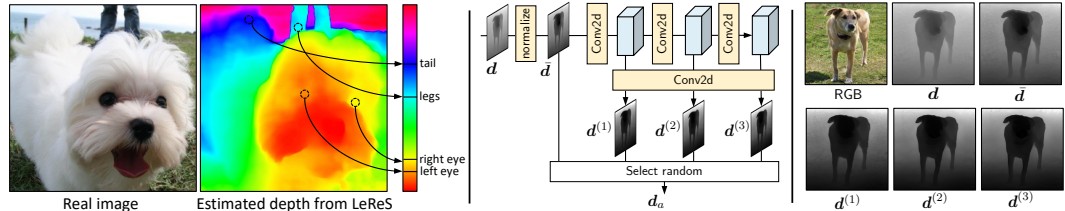

Figure 4: **Depth adapter.** Left: An example of a real image with its depth estimated by LeReS (Miangoleh et al., 2021). Note that the estimated depth has several artifacts. For example, the human legs are closer than the tail, eyes are spaced unrealistically, and far-away grass is predicted to be close. Middle: depth adapter meant to bridge the domains of predicted and NeRF-rendered depth. Right: a generated image with its adapted depth maps obtained from different layers of the adapter.

arriving at the following regularization term:

$$\mathcal{L}_{\varphi_i} = \left| \frac{\partial \varphi_i}{\partial \varphi_i'} \right| + \left| \frac{\partial \varphi_i}{\partial \varphi_i'} \right|^{-1}, \tag{1}$$

where $\varphi_i' \in \varphi'$ is the camera sampled from the prior distribution and $\varphi_i \in \varphi$ is produced by the camera generator. We refer to this loss as Camera Gradient Penalty. Its first part prevents rapid camera changes, facilitating stable optimization, while the second part avoids collapsed posteriors.

### 3.2 ADVERSARIAL DEPTH SUPERVISION

To instill a 3D bias into our model, we develop a strategy of using depth maps predicted by an off-the-shelf estimator E (Miangoleh et al., 2021), for its advantages of being generic and readily applicable for many object categories. The main idea is concatenating a depth map as the 4-th channel of the RGB as the input of the discriminator. The fake depth maps in this case is obtained with the help of neural rendering, while the real depth maps are estimated using monocular depth estimator E. However, naively utilizing the depth from E leads to training divergence. This happens because E could only produce relative depth, not metric depth. Moreover, monocular depth estimators are still not perfect, they produce noisy artifacts, ignore high-frequency details, and make prediction mistakes. Thus, we devise a mechanism that allows utilization of the imperfect depth maps. The central part of this mechanism is a learnable depth adaptor A, that should transform and augment the depth map obtained with neural rendering to look like a depth map from E.

More specifically, we first render raw depths $\boldsymbol{d}$ from NeRF via volumetric rendering:

$$\boldsymbol{d} = \int_{t_n}^{t_f} T(t)\sigma(r(t))t dt, \tag{2}$$

where $t_n, t_f \in \mathbb{R}$ are near/far planes, $T(t)$ is accumulated transmittance, and $r(t)$ is a ray. Raw depth is shifted and scaled from the range of $[t_n, t_f]$ into $[-1, 1]$ to obtain normalized depth $\bar{\boldsymbol{d}}$:

$$\bar{\boldsymbol{d}} = 2 \cdot \frac{\boldsymbol{d} - (t_n + t_f + b)/2}{t_f - t_n - b}, \tag{3}$$

where $b \in [0, (t_n + t_f)/2]$ is an additional learnable shift needed to account for the empty space in the front of the camera. Real depths are normalized into the $[-1, 1]$ range directly.

**Learnable Depth Adaptor.** While $\bar{\boldsymbol{d}}$ has the same range as $\boldsymbol{d}_r$, it is not suitable for adversarial supervision directly due it the imprecision of E: G would be trained to simulate all its prediction artifacts. To overcome this issue, we introduce a *depth adaptor* A to adapt the depth map $\boldsymbol{d}_a = A(\bar{\boldsymbol{d}}) \in \mathbb{R}^{h \times w}$, where $h \times w$ is a number of sampled pixels. This depth (fake $\boldsymbol{d}_a$ or real $\boldsymbol{d}_r$) is concatenated with the RGB input and passed to D.

The depth adaptor A models artifacts produced by E, so that the discriminator should focus only on the relevant high level geometry. However, a too powerful A would be able to fake the depth completely, and G will not learn the geometry. This is why we structure A as just a 3-layer convolutional network (see Fig. 4). Each layer produces a separated depth map with different levels

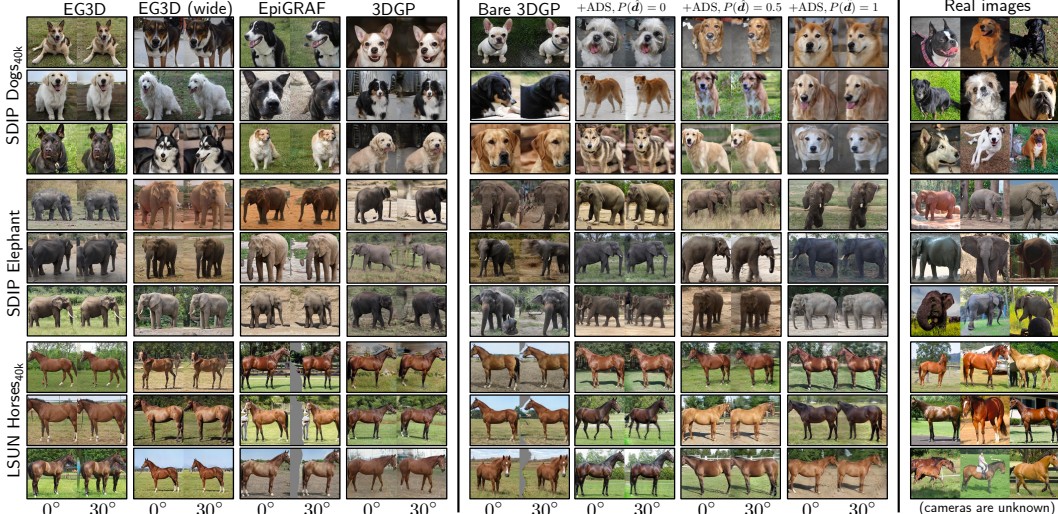

Figure 5: **Qualitative multi-view comparisons.** Left: samples from the models trained on single-category datasets with articulated geometry. Two views are shown for each sample. Middle: ablations following Tab. 1, where we change the probability of using the normalized rendered depth $P(\bar{d})$. EG3D, EpiGRAF, and $P(\bar{d}) = 0$ do not render realistic side views, due to the underlying flat geometry. Our full model instead generates realistic high-quality views on all the datasets. Right: randomly sampled real images. Zoom-in for greater detail.

of adaptation: $\boldsymbol{d}_a^{(1)}, \boldsymbol{d}_a^{(2)}$ and $\boldsymbol{d}_a^{(3)}$. The final adapted depth $\boldsymbol{d}_a$ is randomly selected from the set of $\{\bar{\boldsymbol{d}}, \boldsymbol{d}_a^{(1)}, \boldsymbol{d}_a^{(2)}, \boldsymbol{d}_a^{(3)}\}$. Such design can effectively learn good geometry while alleviating overfitting. For example, when D receives the original depth map $\bar{\boldsymbol{d}}$ as input, it provides to G a strong signal for learning the geometry. And passing an adapted depth map $\boldsymbol{d}_a^{(i)}$ to D allows G to simulate the imprecision artifacts of the depth estimator without degrading its original depth map $\bar{\boldsymbol{d}}$.

### 3.3 KNOWLEDGE DISTILLATION FOR DISCRIMINATOR

Knowledge from pretrained classification networks was shown to improve training stability and generation quality in 2D GANs (Sauer et al., 2021; Kumari et al., 2022; Sauer et al., 2022; Casanova et al., 2021). A popular solution proposed by Sauer et al. (2021; 2022) is to use an off-the-shelf model as a discriminator while freezing most of its weights. Unfortunately, this technique is not applicable in our scenario since we modify the architecture of the discriminator by adding an additional depth input (see §3.2) and condition on the parameters of the patch similarly to EpiGRAF (Skorokhodov et al., 2022). Thus, we devise an alternative technique that can work with arbitrary architectures of the discriminator. Specifically, for each real sample, we obtain two feature representations: $e$ from the pretrained ResNet (He et al., 2016) network and $\hat{e}$ extracted from the final representation of our discriminator D. Our loss simply pushes $\hat{e}$ to $e$ as follows:

$$\mathcal{L}_{\mathrm{dist}} = \|\boldsymbol{e} - \hat{\boldsymbol{e}}\|_2^2. \tag{4}$$

$\mathcal{L}_{\mathrm{dist}}$ can effectively distill knowledge from the pretrained ResNet (He et al., 2016) into our D.

### 3.4 TRAINING

The overall loss for generator G consists of two parts: adversarial loss and Camera Gradient Penalty:

$$\mathcal{L}_{\mathrm{G}} = \mathcal{L}_{\mathrm{adv}} + \sum_{\varphi_i \in \boldsymbol{\varphi}} \lambda_{\varphi_i} \mathcal{L}_{\varphi_i}, \tag{5}$$

where $\mathcal{L}_{\mathrm{adv}}$ is the non-saturating loss (Goodfellow et al., 2014). We observe that a diverse distribution for camera origin is most important for leaning meaningful geometry, but it is also most prone to degrade to a constant solution. Therefore, we set $\lambda_{\varphi_i} = 0.3$ for $\boldsymbol{\varphi}_{\mathrm{pos}}$, while set $\lambda_{\varphi_i} = 0.03$ for $\varphi_{\mathrm{fov}}$ and $\lambda_{\varphi_i} = 0.003$ for $\varphi_{\mathrm{lookat}}$. The loss for discriminator D, on the other hand, consists of three

parts: adversarial loss, knowledge distillation, and $\mathcal{R}_1$ gradient penalty (Mescheder et al., 2018):

$$\mathcal{L}_D = \mathcal{L}_{\text{adv}} + \lambda_{\text{dist}}\mathcal{L}_{\text{dist}} + \lambda_r \mathcal{R}_1. \qquad (6)$$

We use the same optimizer and hyper-parameters as EpiGRAF. We observe that for depth adaptor, sampling adapted depth maps with equal probability is not always beneficial, and found that using $P(\bar{d}) = 0.5$ leads to better geometry. For additional details, see Appx B.

## 4 EXPERIMENTAL RESULTS

**Datasets.** In our experiments, we use 4 *non-aligned* datasets: SDIP Dogs (Mokady et al., 2022), SDIP Elephants (Mokady et al., 2022), LSUN Horses (Yu et al., 2015), and ImageNet (Deng et al., 2009). The first three are single-category datasets that contain objects with complex articulated geometry, making them challenging for standard 3D generators. We found it useful to remove outlier images from SDIP Dogs and LSUN Horsesvia instance selection (DeVries et al., 2020), reducing their size to 40K samples each. We refer to the filtered versions of these datasets as SDIP Dogs$_{40k}$, and LSUN Horses$_{40k}$, respectively. We then validate our method on ImageNet, a real-world, multi-category dataset containing $1,000$ diverse object classes, with more than $1,000$ images per category. All the 3D generators (including the baselines) use the same filtering strategy for ImageNet, where 2/3 of its images are filtered out. Note that all the metrics are *always* measured on the full ImageNet.

**Evaluation.** We rely on FID (Heusel et al., 2017) to measure the image quality and FID$_{2k}$, which is computed on 2,048 images instead of 50k (as for FID) for efficiency. For ImageNet, we additionally compute Inception Score (IS) (Salimans et al., 2016). Note that while we train on the filtered ImageNet, we always compute the metrics on the full one. There is no established protocol to evaluate the geometry quality of 3D generators in general, but the state-of-the-art ones are tri-plane (Chan et al., 2022; Sun et al., 2022) or MPI-based Zhao et al. (2022), and we observed that it is possible to quantify their most popular geometry failure case: flatness of the shapes. For this, we propose Non-Flatness Score (NFS) which is computed as the average entropy of the normalized depth maps his-

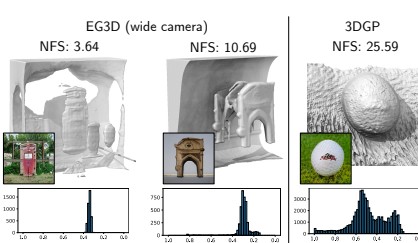

Figure 6: NFS for repetitive and diverse geometry with depth histograms.

tograms. We depict its intuition in Fig 6 and provide the details in Appx F.

### 4.1 3D GENERATION FOR SINGLE CATEGORY DATASETS

We show quantitative results for single category datasets in Tab. 1a, where we compare with EG3D (Chan et al., 2022) and EpiGRAF (Skorokhodov et al., 2022). EG3D was mainly designed for FFHQ (Karras et al., 2019) and uses the true camera poses inferred from real images as the camera distribution for the generator. But in our scenario, we do not have any knowledge about the true camera distribution, that is why to stay as close as possible to the setup EG3D was designed for, we use normal distribution for rotation and elevation angles with the same standard deviations as in FFHQ, which are equal to $\sigma_{\text{yaw}} = 0.3$ and $\sigma_{\text{pitch}} = 0.155$, respectively. Also, to learn better geometry, we additionally trained the baselines with a twice wider camera distribution: $\sigma_{\text{yaw}} = 0.6$ and $\sigma_{\text{pitch}} = 0.3$. While it indeed helped to reduce flatness, it also worsened the image quality: up to 500% as measured by FID$_{2k}$. Our model shows substantially better (at least $2\times$ than EG3D, slightly worse than StyleGAN2) FID$_{2k}$ and greater NFS on all the datasets. Low NFS indicates flat or repetitive geometry impairing the ability of the model to generate realistic side views. Indeed, as shown in Fig. 5 (left), both EG3D (with the default camera distribution) and EpiGRAF struggle to generate side views, while our method (3DGP) renders realistic side views on all three datasets.

**Adversarial Depth Supervision.** We evaluate the proposed Adversarial Depth Supervision (ADS) and the depth adaptor A. The only hyperparameter it has is the probability of using the non-adapted depth $P(\bar{d})$ (see §3.2). We ablate this parameter in Tab. 1b. While FID scores are only slightly affected by varying $P(\bar{d})$, we see substantial difference in the Non-Flatness Score. We first verify that NFS is the worst without ADS, indicating the lack of a 3D bias. When $P(\bar{d}) = 0$, the discriminator is never presented with the rendered depth $\bar{d}$, while the adaptor learns to fake the depth,

Table 1: Comparisons on SDIP Dogs$_{40k}$, SDIP Elephants, and LSUN Horses$_{40k}$. (a) includes EG3D (with the standard and the wider camera range), EpiGRAF, and 3DGP. For completeness, we also provide 2D baseline StyleGAN2 (with KD). (b) includes the ablations of the proposed contributions. We report the total training cost for prior works, our model, and our proposed contributions.

(a) Comparison of EG3D, EpiGRAF and 3DGP.

| Model | SDIP Dogs$_{40k}$ | | SDIP Elephants$_{40k}$ | | LSUN Horses$_{40k}$ | | Training cost (A100 days) |
|---|---|---|---|---|---|---|---|
| | FID$_{2k}$ ↓ | NFS↑ | FID$_{2k}$ ↓ | NFS↑ | FID$_{2k}$ ↓ | NFS↑ | |
| EG3D | 16.2 | 11.91 | 4.78 | 2.59 | 3.12 | 13.34 | 3.7 |
| + wide camera | 21.1 | 24.44 | 5.76 | 17.88 | 19.44 | 25.34 | 3.7 |
| EpiGRAF | 25.6 | 3.53 | 8.24 | 12.9 | 6.45 | 9.73 | 2.3 |
| 3DGP (ours) | 8.74 | 34.35 | 5.79 | 32.8 | 4.86 | 30.4 | 2.6 |
| StyleGAN2 (with KD) | 6.24 | N/A | 3.94 | N/A | 2.57 | N/A | 1.5 |

(b) Impact of Adversarial Depth Supervision (ADS).

| | | | | | | | |
|---|---|---|---|---|---|---|---|
| Bare 3DGP (w/o ADS, w/o C) | 8.59 | 1.42 | 7.46 | 9.52 | 3.29 | 8.04 | 2.3 |
| + ADS, $P(\bar{d}) = 0.0$ | 8.13 | 3.14 | 5.69 | 1.97 | 3.41 | 1.24 | 2.6 |
| + ADS, $P(\bar{d}) = 0.25$ | 9.57 | 33.21 | 6.26 | 33.5 | 4.33 | 32.68 | 2.6 |
| + ADS, $P(\bar{d}) = 0.5$ | 9.25 | 36.9 | 7.60 | 30.7 | 5.27 | 32.24 | 2.6 |
| + ADS, $P(\bar{d}) = 1.0$ | 12.2 | 27.2 | 12.1 | 26.0 | 8.24 | 27.7 | 2.5 |

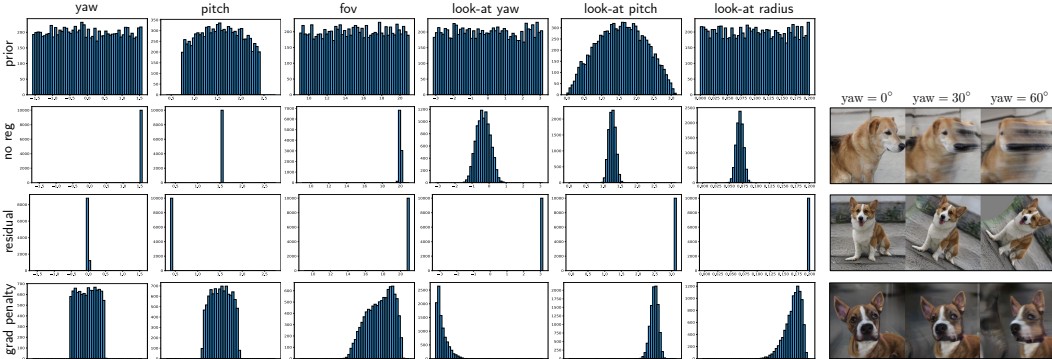

Figure 7: **Comparisons of regularization strategies.** Left: selected generic and wide prior for each of the 6 DoFs of our camera model (top row). Without any regularization (*no reg*) or with the residual-based model (*residual*) as in Niemeyer & Geiger (2021a), the camera generator collapses to highly concentrated distributions. In contrast, the proposed regularization (*grad penalty*) leads to a wider posterior. Right: random samples for each strategy. For (*no reg*) and (*residual*), no meaningful geometry is learned; only our method (*grad penalty*) leads to good geometry.

leading to flat geometry. When $P(\bar{d}) = 1$, the adaptor is never used, allowing D to easily determine the generated depth from the estimated depth, as there is a large domain gap (see Fig. 4), leading to reduced FID scores. The best results are achieved with $P(\bar{d}) = 0.5$, which can be visually verified from observing Fig. 5 (middle). A "bare" 3DGP, $P(\bar{d}) = 0$ and $P(\bar{d}) = 1$ are unable to render side views, while the model trained with $P(\bar{d}) = 0.5$ features the overall best geometry and side views.

**Knowledge Distillation.** Here we further provide insights for discriminator D using our knowledge distillation strategy (see §3.3). We find that knowledge distillation provides an additional stability for adversarial training, along with the significant improvement in FID, which can be observed by comparing results of EpiGRAF in Tab. 1a and Bare 3DGP Tab. 1a. Additionally, we compare different knowledge distillation strategies in Appx C. However, as noted by Parmar et al. (2021), strategies which utilize additional classification networks, may provide a significant boost to FID, without corresponding improvement in visual quality.

**"Ball-in-Sphere" Camera Distribution.** In Fig. 7, we analyze different strategies for learning the camera distribution: sampling the camera $\varphi$ from the prior $\varphi'$ without learning, predicting the residual $\varphi - \varphi'$, and using our proposed camera generator C with Camera Gradient Penalty. For the first two cases, the learned distributions are nearly deterministic. Not surprisingly, no meaningful side views can be generated as geometry becomes flat. In contrast, our regularization provides

Table 2: Comparison between different generators on ImageNet $256^2$ (note that 3DGP relies on additional information in the form of depth supervision). Training cost is measured in A100 days.

| Method | Synthesis type | FID ↓ | IS ↑ | NFS↑ | A100 days ↓ |
|---|---|---|---|---|---|
| BigGAN (Brock et al., 2018) | 2D | 8.7 | 142.3 | N/A | 60 |
| StyleGAN-XL (Sauer et al., 2022) | 2D | 2.30 | 265.1 | N/A | 163+ |
| ADM (Dhariwal & Nichol, 2021) | 2D | 4.59 | 186.7 | N/A | 458 |
| EG3D (Chan et al., 2022) | 3D-aware | 26.7 | 61.4 | 3.70 | 18.7 |
| + wide camera | 3D-aware | 25.6 | 57.3 | 9.83 | 18.7 |
| VolumeGAN (Xu et al., 2021) | 3D-aware | 77.68 | 19.56 | 22.69 | 15.17 |
| StyleNeRF (Gu et al., 2022) | 3D-aware | 56.54 | 21.80 | 16.04 | 20.55 |
| StyleGAN-XL + 3DPhoto (Shih et al., 2020) | 3D-aware | 116.9 | 9.47 | N/A | 165+ |
| EpiGRAF (Skorokhodov et al., 2022) | 3D | 47.56 | 26.68 | 3.93 | 15.9 |
| + wide camera | 3D | 58.17 | 20.36 | 12.89 | 15.9 |
| 3DGP (ours) | 3D | 19.71 | 124.8 | 18.49 | 28 |

sufficient regularization to C, enabling it to converge to a sufficiently wide posterior, resulting into valid geometry and realistic side views. After the submission, we found a simpler and more flexible camera regularization strategy through entropy maximization, which we discuss in Appx J.

## 4.2 3D SYNTHESIS ON IMAGENET

ImageNet (Deng et al., 2009) is significantly more difficult than single-category datasets. Following prior works, we trained *all* the methods in the conditional generation setting Brock et al. (2018). The quantitative results are presented in Tab. 2. We also report the results of state-of-the-art 2D generators for reference: as expected, they show better FID and IS than 3D generators, since they do not need to learn geometry, are trained on larger compute, and had been studied for much longer. Despite our best efforts to find a reasonable camera distribution, both EG3D and EpiGRAF produce flat or repetitive geometry, while 3DGP produces geometry with rich details (see Fig. 1). StyleN-eRF (Gu et al., 2022) and VolumeGAN Xu et al. (2021), trained for conditional ImageNet generation with the narrow camera distribution, substantially under-performed in terms of visual quality. We hypothesize that the reason lies in their MLP/voxel-based NeRF backbones: they have a better 3D prior than tri-planes, but are considerably more expensive, which, in turn, requires sacrifices in terms of the generator's expressivity.

Training a 3D generator from scratch is not the only way to achieve 3D-aware synthesis: one can lift an existing 2D generator into 3D using the techniques from 3D photo "inpainting". To test this idea, we generate 10k images with StyleGAN-XL (the current SotA on ImageNet) and then run 3DPhoto (Shih et al., 2020) on them.[1] This method first lifts a photo into 3D via a pre-trained depth estimator and then inpaints the holes with a separately trained GAN model to generate novel views. This method works well for negligible camera variations, but starts diverging when the camera moves more than $10°$. In Tab. 2, we report its FID/IS scores with the narrow camera distribution: $\sigma_{yaw} \sim \mathcal{N}(0, 0.3)$ and $\sigma_{pitch} \sim \mathcal{N}(\pi/2, 0.15)$). The details are in Appx I.

## 5 CONCLUSION

In this work, we present the first 3D synthesis framework for in-the-wild, multi-category datasets, such as ImageNet. We demonstrate how to utilize generic priors in the form of (*imprecise*) monocular depth and latent feature representation to improve the visual quality and guide the geometry. Moreover, we propose the new "Ball-in-Sphere" camera model with a novel regularization scheme that enables learning meaningful camera distribution. On the other hand, our work still has several limitations, such as sticking background, lower visual quality compared to 2D generators, and no reliable quantitative measure of generated geometry. Additional discussion is provided in Appx A. This project consumed $\approx$12 NVidia A100 GPU years in total.

---

[1]We computed FID on 10k images rather than 50k (as for other generators) due to the computational costs: parallelized inference of 3DPhoto (Shih et al., 2020) on 10k images took 2 days on 8 V100s.

## 6 REPRODUCIBILITY STATEMENT

Most importantly, we will release 1) the source code and the checkpoints of our generator as a separate github repo; and 2) fully pre-processed datasets used in our work (with the corresponding extracted depth maps). In §3 and §4 and Appx B, and also in the figures throughout the text, we provided a complete list of architecture and optimization details needed to reproduce our results. We are also open to provide any further details on our work in public and/or private correspondence.

## 7 ETHICS STATEMENT

There are two main ethical concerns around deep learning projects on synthesis: generation of fake content to mislead people (e.g. fake news or deepfakes[2]) and potential licensing, authorship and distribution abuse. This was recently discussed in the research community in the context of 2D image generation (by Stable Diffusion Rombach et al. (2022) and DALL-E Ramesh et al. (2022)) and code generation (by Github Copilot[3]). While the current synthesis quality of our generator would still need to be improved to fool an attentive human observer, we still encourage the research community to discuss ideas and mechanisms for preventing abuse in the future.

---

[2]https://en.wikipedia.org/wiki/Deepfake.
[3]https://github.com/features/copilot.

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

## A  Limitations

**Lower visual quality compared to 2D generators**. Despite providing a more reasonable representation of the underlining scene, 3D generators still have a lower visual quality compared to 2D generators. Closing this gap is essential for a wide adaptation of 3D generators.

**Background sticking**. One common 3D artifact of 3DGP is gluing of the foreground and the background. In other words our model predicts a single shape for both and there is no clear separation between the two. One potential cause of this artifact is the dataset bias, where most of the photos are frontal, thus it is not beneficial for the model to explore backward views. Another reason is the bias of tri-planes toward flat geometry. However, all our attempts to replace tri-planes with an MLP-based NeRF led to much worse results (see Appx D). Inventing a different efficient parametrization may be an important direction.

**No reliable quantitative measure of generated geometry**. In this work we introduce Non-Flatness Score as a proxy metric for evaluating the quality of the underlying geometry. However it can capture only a single failure case, specific for generators based on tri-planes: flatness of geometry. Devising a reasonable metric applicable to a variety of scenarios could significantly speed up the progress in this area.

**Camera generator C does not learn fine-grained control**. While our camera generator is conditioned on the class label $c$, and, in theory, it should be able to perform fine-grained control over the class focal length distributions (which is natural since landscape panoramas and close-up view of a coffee mug typically have different focal lengths), it does not do this, as shown in Fig. 8. We attribute this problem to the implicit bias of G to produce large-FoV images due to tri-planes parametrization. Tri-planes define a limited volume box in space, and close-up renderings with large focal length would utilize fewer tri-plane features, hence using less generator's capacity. This is why 3DGP attempts to perform modeling with larger field-of-view values.

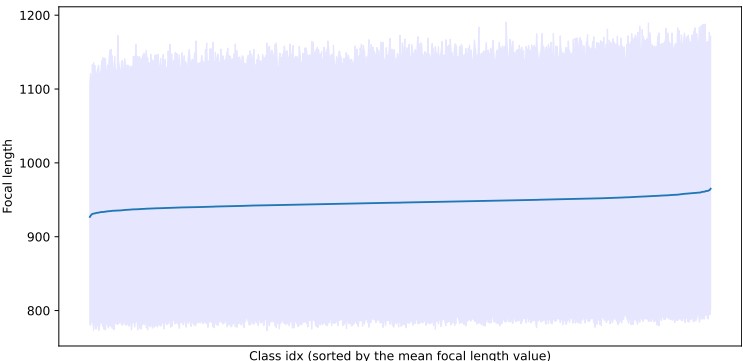

Figure 8: Focal length distribution on ImageNet $256^2$ learned by C. The blue solid line is the mean values, while lower/upper curves are 0.05 and 0.95 quantiles, respectively.

## B  Implementation details

This section provides additional architectural and training details. Also, note that we release the source code. Our generator G consists of M, S and T. Mapping network M takes noise $z \in \mathbb{R}^{512}$ and class label $c \in 0, ..., K - 1$, where K is the number of classes, and produces the style code $w \in \mathbb{R}^{512}$. Similar to StyleGAN2-ADA Karras et al. (2020a) and EpiGRAF Skorokhodov et al. (2022), M is 2-layer MLP network with LeakyReLU activations and 512 neurons in each layer. Synthesis network is a decoder network same as StyleGAN2 Karras et al. (2020b), except that it produce tri-plane feature features $p = (p^{xy}, p^{yz}, p^{xz}) \in \mathbb{R}^{3 \times (512 \times 512 \times 32)}$. A feature vector $f_{xyz} \in \mathbb{R}^{32}$ located $(x, y, z) \in \mathbb{R}^3$ is computed by projecting the coordinate back to the tri-plane representation, followed by bi-linearly interpolating the nearby features and averaging the features from different planes. Following EpiGRAF (Skorokhodov et al., 2022), tri-plane decoder is a two-layer MLP with Leaky-ReLU activations and 64 neurons in the hidden layer, that takes a tri-plane

Table 3: Module names glossary.

| | |
|---|---|
| G | Generator |
| M | Mapping network |
| S | Synthesis network |
| T | Tri-plane decoder |
| C | Camera generator |
| A | Depth adaptor |
| D | Discriminator |
| E | Depth Estimator |
| F | Feature Extractor |

feature $\boldsymbol{f}_{xyz}$ as input and produced the color and density $(\text{RGB}, \sigma)$ in that point. We use same volume rendering procedure as EpiGRAF (Skorokhodov et al., 2022). We also found that increasing the half-life of the exponential moving average for our G improves both FID and Inception Score. We observed this by noticing that the samples change too rapidly over the course of training. In practice, we log the samples every 400k seen images and noticed that the generator could completely change the global structure of a sample during that period.

**Camera generator**. Camera generator consists of linear layers with SoftPlus activations, it architecture is depicted in Fig. 9. We found it crucial to use SoftPlus activation and not LeakyReLU, since optimization of Camera Gradient Penalty for non-smooth functions is unstable for small learning rates (smaller 0.02). Our gradient penalty minimizes the function $\mathcal{L} = |g| + 1/|g|$, where $|g|$ is the input/output scalar derivative of the camera generator C. The motivation is to prevent the collapse of C into delta distribution and the intuition is the following. C can collapse into delta distribution in two ways: 1) by starting to produce the constant output for all the input values (this is being prevented by the first term $|g|$ and 2) by starting producing $\pm\infty$ for all the inputs, which are at the end converted to constants since we apply sigmoid normalization on top of its outputs to normalize them into a proper range (e.g., pitch is bounded in $(0, \pi)$) — this, in turn, is prevented by the second term $1/|g|$. The minimum value of this function is 2 (which is achieved when the gradient norm is constant and equals to 1), and the function itself is visualized in Fig. 9b.

**Adversarial depth supervision**. For depth adaptor, we use a three layer convolutional neural network with $5\times5$ kernel sizes (since it is shallow, we increase its receptive field by increasing the kernel size) with LeakyReLU activations and 64 filters in each layer. Additionally we use one shared convolutional layer that converts $64 \times h \times w$ features to the depth maps. We use the same architecture for D as EpiGRAF (Skorokhodov et al., 2022), but additionally concatenate a 1-channel depth to the 3-channel RGB input. Finally E and F is pretrainined LeReS (Miangoleh et al., 2021) and ResNet50 (He et al., 2016) networks without any modifications. We used the `timm` library to extract the features for real images.

Similarly to EpiGRAF (Skorokhodov et al., 2022) we set the $\mathcal{R}_1$ regularization (Mescheder et al., 2018) term $\lambda_r = 0.1$ and knowledge distillation term $\lambda_{\text{dist}} = 1$. $\mathcal{R}_1$ regularization helps to stabilize GAN training and is formulated as a gradient penalty on top of the discriminator's real inputs:

$$\mathcal{R}_1 = \frac{1}{2}\|\nabla_{\boldsymbol{x}}\mathsf{D}(\boldsymbol{x})\|_2^2 \longrightarrow \min_{\mathsf{D}}. \tag{7}$$

The (hand-wavy) intuition is that it makes the discriminator surface more flat in the vicinity of real points, making it easier for the generator to reach them. We train all the models with Adam optimizer (Kingma & Ba, 2014) using the learning rate of $2e$-3 and $\beta_1 = 0.0, \beta_2 = 0.99$. Following EpiGRAF (Skorokhodov et al., 2022), our model uses patch-wise training with $64 \times 64$-resolution patches and uses their proposed $\beta$ scale sampling strategy without any modifications. We use the batch size of 64 in all the experiments, since in early experiments we didn't find any improvements from using a large batch size neither for our model nor for StyleGAN2, as observed by Brock et al. (2018) and Sauer et al. (2022).

As being said in Section 4, we use the instance selection technique by DeVries et al. (2020) to remove image outliers from the datasets since they might negatively affect the geometry. This procedure works by first extracting a 2,048-dimensional feature vector for each image, then fitting a multivariate gaussian distribution on the obtained dataset and removing the images, which features

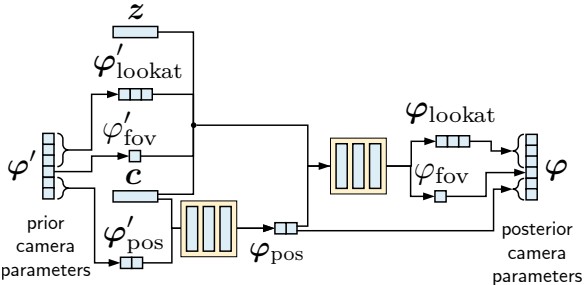
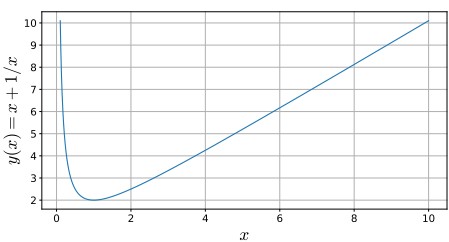

(a) **Camera generator** C**.** We condition C on class labels $c$ when generating the camera position $\hat{\varphi}$ since it might be different for different classes. And we condition it on $z$ when generating the look-at position and field-of-view since it might depend on the object shape (e.g., there is a higher probability to synthesize a close-up view of a dog's snout rather than its tail). Each MLP consists on 3 layers with Softplus non-linearities.

(b) **Camera gradient penalty**. We structure the regularization term for C as a function $\mathcal{L}(|g|) = |g| + 1/|g|$ (see (1)), and this function is visualized above. This allows to prevent the collapse of C into delta distribution by it either producing constant values or very large/small values (which become constant after the sigmoid normalization).

Figure 9: Camera generator architecture and visualizing its corresponding regularization term.

have low probability density. For SDIP Dogs and LSUN Horses, we fit a multi-variate gaussian distribution for the whole dataset. For ImageNet, we fit a separate model for each class with additional diagonal regularization for covariance, which is needed due to its singularity: feature vector has more dimensions than the number of images in a class. We refer to (DeVries et al., 2020) for additional details.

Also note that we will release the source code and the checkpoints, which would additionally convey all other implementation details.

## C  KNOWLEDGE DISTILLATION

In this section we compare our knowledge distillation strategy with projected D (Sauer et al., 2021; 2022), another popular technique of utilizing existing classification models for GAN training. Note that projected D relies on pretrained EfficientNet (Tan & Le, 2019), it freezes most of the weights and adds some random convolutions layers that project intermediate outputs of EfficientNet to the features, that will be later processed by discriminator. Note that this strategy relies on specific architecture of Discriminator D, thus in order to adapt it for 3DGP we have to rely on two discriminators. First one is our disciminator D with additional depth input and conditioned on patch parameters and second one is projected D. In Tab 4 we show comparison between projected D and our knowledge distillation strategy on ImageNet (Deng et al., 2009) and SDIP Dogs$_{40k}$ Mokady et al. (2022) datasets. First we compare these strategies for 2D generator architectures: StyleGAN3-t (large) (Karras et al., 2021) and StyleGAN2 (Karras et al., 2020a). It can be observed that projected D is not generic and it results heavily varies depending on generator architecture. Moreover training cost of projective D is higher. Finally we test this strategy for our 3D generator, and again we can observe that this technique lead to inferior results in both visual quality and training cost.

Typically, knowledge distillation in classifiers is performed using Kullback-Leibler divergence on top of logits (Hinton et al., 2015; Kim et al., 2021), rather than $\mathcal{L}_2$ distance on top of hidden activations, as in our case. There are two design reasons of why we use the $\mathcal{L}_2$: 1) it is more generalizable and one can transfer knowledge from other models with the same design, like CLIP (Radford et al., 2021) or Mask R-CNN (He et al., 2017); 2) our discriminator's task shouldn't be able to perform classification, that is why guiding it with the KL classification loss is not natural. However, in Fig. 10, we provide additional exploration with other knowledge distillation objectives on top of StyleGAN2 (Karras et al., 2020b) trained for conditional ImageNet generation on the $128^2$ resolution. One can observe that KL and $\mathcal{L}_2$ objectives perform approximately the same. Also, guiding by CLIP underperforms to ResNet guidance initially, but starts to outperform after more training is performed: we hypothesize that the reason is that it is a more general model. We do not use CLIP

Table 4: Comparing our knowledge transfer strategy with the one from StyleGAN-XL.

| Method | ImageNet $128^2$ @ 10M | | SDIP Dogs$_{40k}$ $256^2$ @ 5M | | Restricts D's |
| | FID↓ | Training cost ↓ | FID ↓ | Training cost ↓ | architecture? |
| --- | --- | --- | --- | --- | --- |
| StyleGAN3-t (large) | 28.1 | 11.9 | 6.42 | 7.3 | No |
| - with Projected D | 22.8 | 11.6 | 22.6 | 8.1 | Yes |
| - with Knowledge Distillation | 16.3 | 11.9 | 2.47 | 7.3 | No |
| StyleGAN2 | 33.7 | 1.81 | 9.79 | 1.3 | No |
| - with Projected D | 160.5 | 3.83 | 4.10 | 2.1 | Yes |
| - with Knowledge Distillation with $\mathcal{L}_2$ | 20.75 | 1.83 | 2.08 | 1.4 | No |
| - with Knowledge Distillation with KL | 25.92 | 1.83 | 2.54 | 1.4 | No |
| 3DGP | 53.6 | 6.33 | 21.3 | 2.6 | No |
| - with Projected D | 105.9 | 8.0 | 10.5 | 4.2 | Yes |
| - with Knowledge Distillation | 27.8 | 6.83 | 4.51 | 2.6 | No |

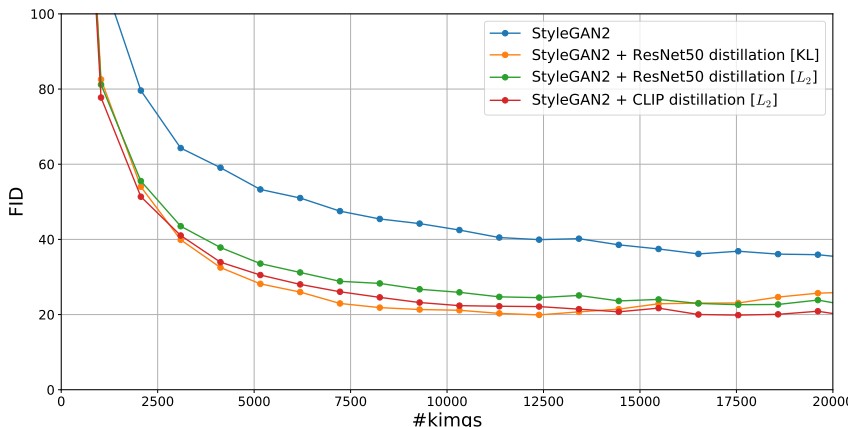

Figure 10: **Exploring different knowledge distillation objectives**. Each model is trained for conditional ImageNet generation on the $128^2$ resolution with all other hyperparameters being the same.

guidance for our generators in this paper so not to give it an unfair advantage compared to other models.

# D  FAILED EXPERIMENTS

In this section we describe several ideas that have been tried in this project, which however did not work for us.

**Gradient Penalty for Depth Adaptor**. To prevent depth adaptor A from completely faking the depth and forcing it to still provide a useful signal to the generator, one could employ a gradient penalty that will force gradient of A close to one. We tried it in two setups: window-to-pixel gradient and pixel-to-pixel gradient. Even with the large weight for this loss and shallow adaptor with $k = 3$ kernel size, it didn't enforce good geometry

**3DGP with Projected** D. Before devoting to knowledge distillation, we spend significant resources trying to incorporate projected D in our setup. For double discriminators setting, see Appx C, we tested: different weighting strategies, less frequent updates for projected D, enabling projected D after some geometry was already learned. We also tested single projected D, where we learned additional depth encoder that can consume our depth inputs. In all these experiments learned geometry was significantly inferior to the setting without projected D. Most of the time geometry becomes completely flat.

**MLPs instead of tri-planes**. We noticed that tri-planes are extremely biased towards flat generation. Therefore one natural idea would be to utilize different representations, one possible candidate is MLPs Chan et al. (2021). However our experiments with MLPs always lead to significantly inferior

visual quality. Our hypothesis is that they just don't have enough capacity to model large scale datasets.

**Few-shot NeRF regularization**. We also try to improve learned geometry by incorporating ray entropy minimization loss from InfoNeRF Kim et al. (2022) with different weights. But the model always diverges. We hypothesize that this is because it is significantly more complicated to find proper geometry, if the model is stuck in local minimum where the object densities are very sharp.

**Normal supervision**. Another prominent idea was utilizing normal supervision, since generic networks for normals Eftekhar et al. (2021) are also widely available and provide good results on the arbitrary data. Since computing normals requires gradient of density $\sigma$ with respect to $(x, y, z)$ and pytorch does not provide second order derivatives for `F.grid_sample` [4], we developed a custom CUDA kernel that computes second derivative for `F.grid_sample`. Unfortunately we found that normals supervision is inferior to depth and it is much easier for the generator G to fake normals.

**Preventing C collapse via variance/entropy/moments regularization**. Before arriving to our current form we had been experimenting with other forms of regularization: maximizing variance with some small loss coefficient, or entropy (in the assumption that the posterior distribution is gaussion), or additionally pushing mean/skewness/kurtosis to the one of the gaussian distribution. But each time, the generator was finding the ways to "cheat" the regularization and managing to produce either a delta distribution or a mixture of delta distributions (it very "likes" doing so since, it is able to completely flat images and cheat the geometry in this regime).

## E  ADDITIONAL SAMPLES

Since it is much easier to visualize the samples from NeRF-based generators as videos rather than RGB images, we provide all the additional visualizations on https://snap-research.github.io/3dgp.

## F  NON-FLATNESS SCORE DETAILS

To detect and quantify the flatness of 3D generators, we propose a Non-Flatness Score (NFS). To compute NFS, we sample $N$ latent codes with their corresponding radiance fields. For each radiance field, we perform integration following Eq. 2 to obtain its depth, however, we first set the $50\%$ of lowest density to zero. This is necessary to cull spurious density artifacts.

We then normalize each depth map according to the corresponding near and far planes and compute a histogram with $B$ bins, showing the distribution of the depth values. To analyze how much the depth values are concentrated versus spread across the volume, we compute its entropy for each distribution. Averaging over $N$ depth maps gives the sought score:

$$\text{NFS} = \frac{1}{N} \sum_{i=1}^{N} \exp\left[ -\frac{1}{h \cdot w} \sum_{j=1}^{B} b(\boldsymbol{d}^{(i)})_j \right], \tag{8}$$

where $b(\boldsymbol{d}^{(i)})_j$ is the normalized number of depth values in the $j$-th bin of the $i$-th depth map $\boldsymbol{d}^{(i)}$, with $N = 256$ and $B = 64$. NFS does not directly evaluate the quality of the geometry. Instead, it helps to detect and quantify how flat the generated geometry is. In Fig. 6, we show examples of repetitive geometry generated by EG3D and a more diverse generation produced by 3DGP along with their depth histograms and NFS.

Intuitively, NFS should provide high values to EG3D with a wide camera distribution, since its repetitive shapes are supposed to span the whole ray uniformly. But surprisingly, it does not do so and is able to relfect this repetitiveness artifact as well, yielding low scores, as can be observed from Tables 1 and 2.

## G  INVESTIGATING THE POTENTIAL DATA LEAK FROM DEPTH ESTIMATOR

LeReS (Miangoleh et al., 2021) depth estimator (which was used in our work) was pre-trained on a different set of images compared to ImageNet (or other animals datasets, explored in our work),

---

[4]https://github.com/pytorch/pytorch/issues/34704

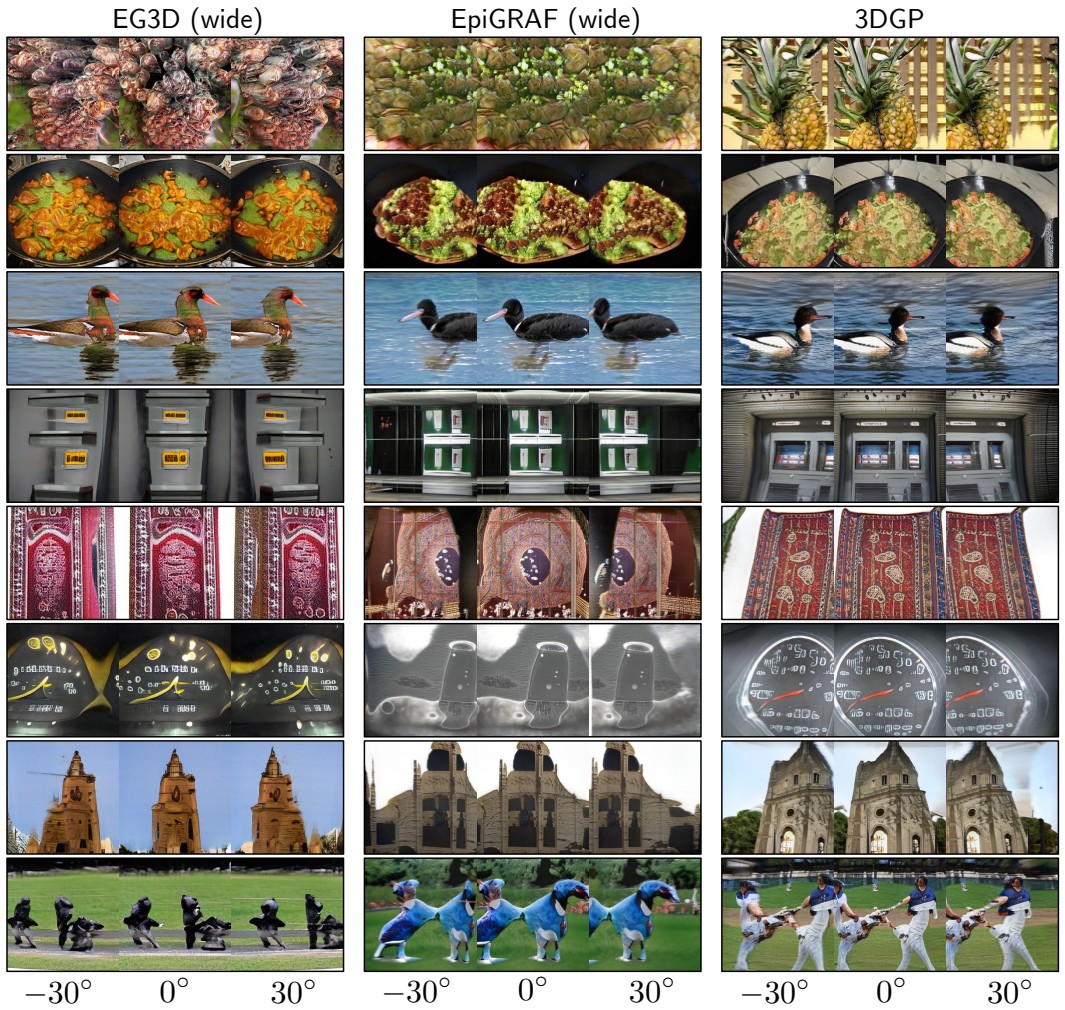

Figure 11: *Random* samples for *random* classes on ImageNet $256^2$ (random seed of 1 for the first ImageNet classes).

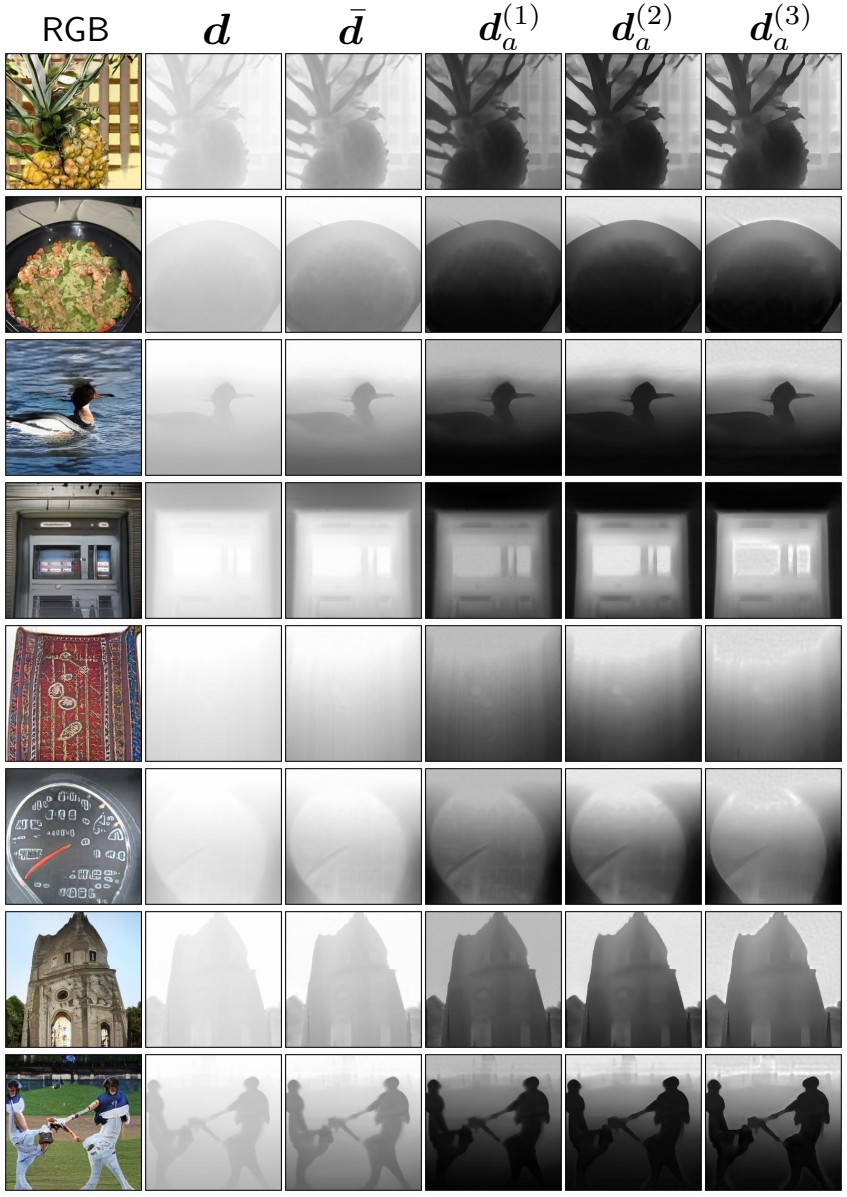

Figure 12: Depth maps produced by 3DGP.

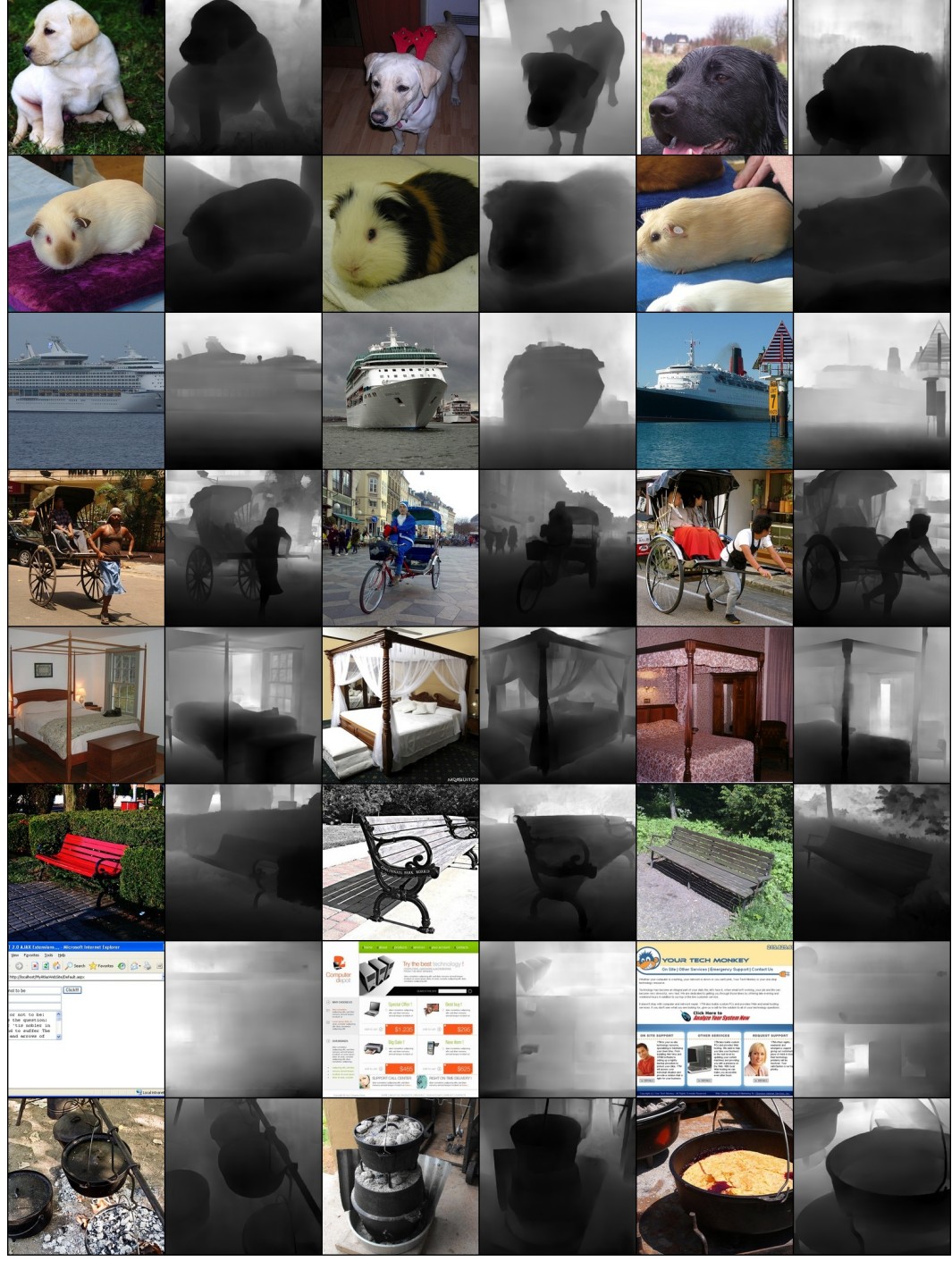

Figure 13: Depth maps on ImageNet $256^2$, predicted by LeReS (Miangoleh et al., 2021) depth estimator. See the generated depth by 3DGP in Fig. 12.

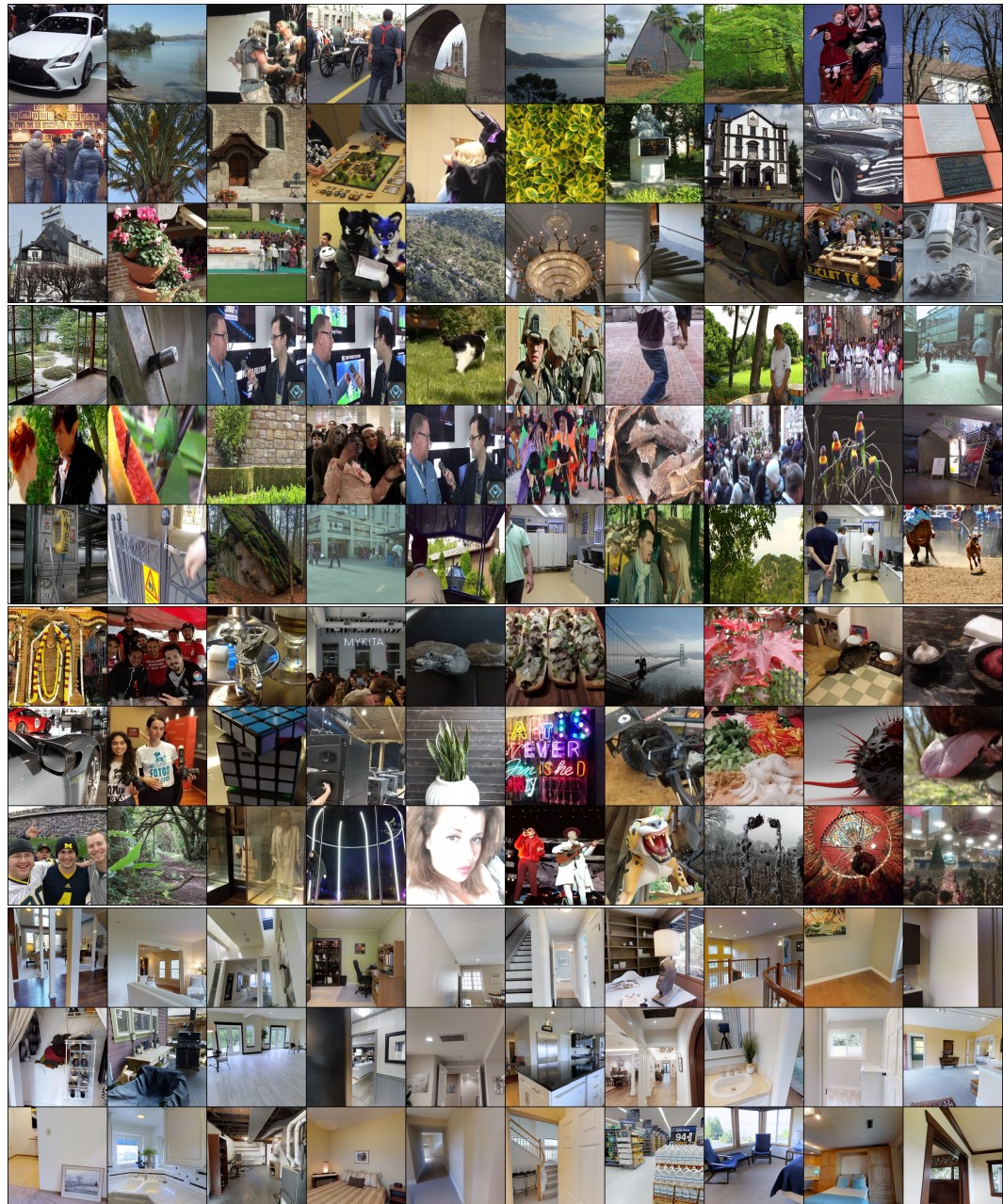

Figure 14: Random images from each pre-training dataset of the LeReS (Miangoleh et al., 2021) depth estimator, used in our work. From top to bottom: HRWSI (Xian et al., 2020), DiverseDepth (Yin et al., 2021), Holopix50k (Hua et al., 2020), and Taskonomy (Zamir et al., 2018).

and this theoretically could help our generator achieve better results by implicitly giving it access to a broader image set rather than giving only texture-less geometric guidance. In this section, we investigate this, and conclude that it is not the case: the pre-training data of LeReS is too different from the explored datasets, containing almost no animal images — while all the explored datasets are very animal-dominated, including ImageNet which contains 482 animal classes. Hence, LeReS has no chance in helping our generator by giving it access to additional data.

LeReS (Miangoleh et al., 2021) trains on a combination of 4 datasets (in fact, parts of them): 1) HRWSI (Xian et al., 2020) contains outdoor city imagery (e.g., buildings, monuments, land-scapes); 2) DiverseDepth (Yin et al., 2021) contains clips from in-the-wild movies and videos; 3)

Holopix50k (Hua et al., 2020) — diverse, in-the-wild web images (it is the most similar to ImageNet in terms of underlying data distribution); and 4) Taskonomy (Zamir et al., 2018) contains indoor scenes (bedrooms, stores, etc.). In Fig. 14, we provide 30 random images from each dataset. In Tab. 6, we rigorously explore how many animal images does the pre-training data of LeReS contains. We perform this in two ways: by directly counting the amount of animals with the pre-trained Mask R-CNN model He et al. (2017) and by computing FID scores between the pre-training datasets of LeReS and ImageNet animal subset. The pre-trained Mask R-CNN model from TorchVision is able to detect 10 animal classes: birds, cats, dogs, horses, sheeps, cows, elephants, bears, zebras and giraffes. Following the official tutorial, we used a threshold of 0.8. We downloaded the datasets with the official download scripts of LeReS.

Table 5: Investigating the distribution overlap between LeReS Miangoleh et al. (2021) training data and the animals subset of ImageNet.

| Dataset | #images | Animal subset | | FID | |
| | | #images | %images | ImageNet | ImageNet$_{animals}$ |
|---|---|---|---|---|---|
| HRWSI Xian et al. (2020) | 18.2k | 1.05k | 5.75% | 71.48 | 97.79 |
| DiverseDepth Yin et al. (2021) | 95.4k | 7.9k | 8.26% | 90.99 | 122.6 |
| Holopix50k Hua et al. (2020) | 42k | 4k | 9.56 | 47.44 | 81.62 |
| Taskonomy Zamir et al. (2018) | 134.7k | 0.5k | 0.36% | 135.1 | 154.3 |
| All pre-training data | 290k | 9.4k | 3.79% | 77.46 | 108.6 |
| ImageNet | 1281.2k | 618.7k | 48.3% | 0.0 | 19.06 |

From Tab. 5, one can see that the pre-training data of LeReS contains just 3.79% of animal images, while ImageNet contains 48.3%. In this way, it is too far away in terms of distribution from the Animal subset of ImageNet. From this, one can conclude the following: *Depth Estimator has almost never seen animals during training, since its training datasets have 600 fewer animal images than ImageNet.* This means that our generator does not receive any unfair advantage in synthesizing animals from using adversarial depth supervision by implicitly getting access to a larger set of images.

How does it affect our generator's performance in synthesizing animals? Will it have an unusually poor FID compared to non-animal data or compared to other methods? In Tab. 6, we report FID scores of each generator on animal vs non-animal subsets of ImageNet. From these scores, one can see that the trend is the same as for FID on full ImageNet. This implies that our generator performs equally well on data which constitutes the main part of our training dataset and is not a part of the depth estimator pre-training.

In this way, one can conclude that adversarial depth supervision helps to achieve better synthesis quality only through geometric guidance. It does not help the generator to do this by leaking the knowledge of pre-trained data from the pre-trained depth estimator.

Table 6: FID scores of different generators on 482 animal classes of ImageNet $256^2$ (each generator was trained on all the classes of ImageNet $256^2$). For this evaluation, we used the images only from the animal classes for both real and fake data.

| Method | Synthesis type | Animal FID ↓ | Non-Animal FID ↓ | FID ↓ |
|---|---|---|---|---|
| StyleGAN-XL | 2D | 4.53 | 4.81 | 2.30 |
| EG3D (wide camera) | 3D-aware | 44.55 | 49.14 | 25.6 |
| EpiGRAF | 3D | 78.65 | 81.38 | 47.56 |
| 3DGP (ours) | 3D | 47.32 | 50.94 | 26.47 |

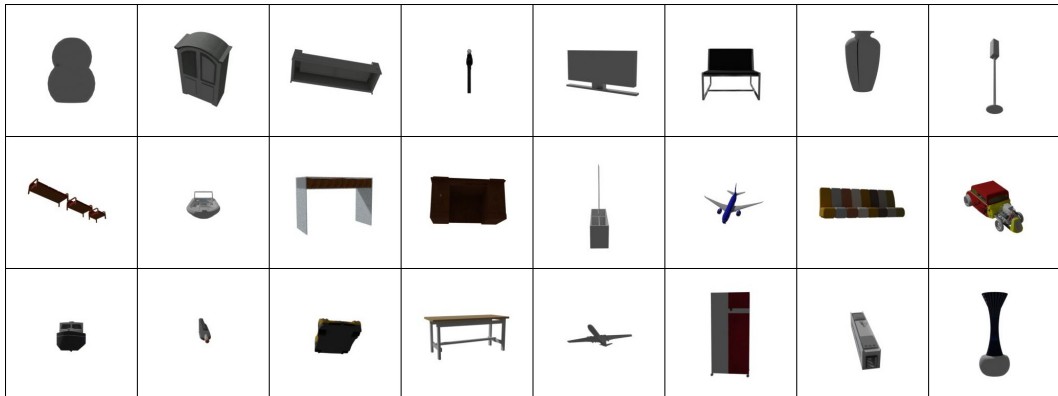

Figure 15: Examples of real images for our rendered ShapeNet $256^2$ dataset.

## H   GEOMETRY EVALUATION ON SHAPENET

In this section, we present the details and the results of an additional study, which rigorously shows that our proposed adversarial depth supervision improves the geometry quality.

### H.1   RENDERING DETAILS

We take the ShapeNet dataset Chang et al. (2015), which consists of  50k models from 54 classes and render it from random *frontal* camera positions. For this, we set the camera on a sphere of radius 2 and randomly choose its position by sampling rotation and elevation angles from $N(0, \pi/8)$ and $N(\pi/2, \pi/8)$, respectively. We chose frontal views to be closer to the real-world scenario: the most popular image synthesis datasets are dominated by the frontal views.  Also, if the dataset is not frontal, then there is less necessity in additional geometry supervision: it is a good enough 3D bias (which does not exist in modern in-the-wild datasets). For wide camera distribution, a generator can learn proper geometry on its own: see experiments on synthetic datasets with 360 degrees camera coverage by Skorokhodov et al. (2022). We render just a single view per model since this is also the scenario which we typically have in real datasets (e.g., in all our explored datasets, there is just a single view per scene). Some of the models in ShapeNet are broken (they lack the corresponding mesh files), so we discard them.  In total, this gave us 51209 training images, which are visualized in Fig. 15. We used BlenderProc Denninger et al. (2019) to do rendering.  Also, during rendering we removed the transparent elements of the meshes since they were providing aliasing issues in the depth maps.

### H.2   EXPERIMENTS

**Experimental setup**. After rendering, we trained our main baselines, EG3D (Chan et al., 2022) and EpiGRAF (Skorokhodov et al., 2022), on this dataset using the ground truth camera parametrization and distribution when sampling the camera poses. For 3DGP, we trained it in the following variants: 1) with the default adversarial depth supervision (i.e., using $P(\bar{d}) = 0.5$); and 2-3) with the adversarial depth supervision, but with corrupted depth maps, where corruption is simulated by blurring, similar to (DeVries et al., 2021).  Also, we trained StyleGAN2 (Karras et al., 2020b) as a lower bound on FID. We disabled camera distribution learning and knowledge distillation for the experiments with 3DGP to avoid unnecessary complications: we investigate only adversarial depth supervision in this study. For all the baselines, we use white background in volumetric rendering.

**Evaluation**. We compare the methods in terms of FID (Heusel et al., 2017) and also Frechet Point-cloud Distance (FPD), proposed by Shu et al. (2019). This metric is an FID analog for point clouds: it uses a pre-trained PointNet (Qi et al., 2017a) to extract the features from point clouds and then computes Frechet distance between real and fake representations sets, approximating their underlying distribution as multi-variate normal one.

Table 7: Geometry evaluation for different models on ShapeNet $256^2$ (Chang et al., 2015).

| Method | Synthesis type | FID $\downarrow$ | FPD $\downarrow$ |
|---|---|---|---|
| EG3D | 3D-aware | 17.22 | 2770.5 |
| EpiGRAF | 3D | 21.58 | 424.0 |
| 3DGP (ours) | 3D | 14.38 | 80.67 |
| with $\sigma_{\text{blur}} = 1$ | 3D | 18.33 | 106.9 |
| with $\sigma_{\text{blur}} = 3$ | 3D | 30.37 | 139.5 |
| with $\sigma_{\text{blur}} = 10$ | 3D | 99.41 | 807.4 |
| StyleGAN2 | 2D | 5.54 | N/A |

We used the official codebase of Shu et al. (2019) to compute the metrics. In their original work, Shu et al. (2019) used a customly trained PointNet (Qi et al., 2017a) to extract the features. But we observed an issue with it: it was sometimes producing unusually high FPD scores in the order of $10^6$, even for real point clouds. This is why we extracted pointclouds features with a pre-trained PointNet++ (Qi et al., 2017b) model from a popular public implementation (Yan, 2019).

To extract point clouds from the models, we first extracted the surfaces from them via marching cubes. For this, we sampled density fields in $256^3$ resolution, and then used the marching cubes implementation of PyMCubes (pmneila, 2015) to extract the surfaces. Following EG3D, we thresholded the surface for marching cubes at the density value of $\sigma = 10$. Our overall pipeline is identical to the original procedure of SDF extraction in the EG3D repo, but simpler in terms of implementation. After that, we extracted point clouds by sampling 2,048 points on the surface for both real and fake meshes using uniform sampling from trimesh Dawson-Haggerty et al. (2019).

## H.3 RESULTS

The results of these experiments are presented in Tab. 7. We also provide random sample examples (seed 1, classes 1-8) in Fig. 16. EG3D (Chan et al., 2022) fails to recover the geometry and generates inverted shapes with hollow geometry. This is highlighted by its very high FPD score. EpiGRAF (Skorokhodov et al., 2022) is able to recover shapes, but it models the white background via an additional sphere, which also damages its FPD. Our method, in contrast, learns proper geometry and correctly drops the background. When the depth maps are corrupted with blurring, it deteriorates both its geometry quality and texture quality, which is also confirmed by the metrics and the provided samples.

## I DETAILS ON THE EXPERIMENTS WITH 3D PHOTO INPAINTING

In this section, we provide the details of our experiments on combining 2D generation with 3D Photo Inpainting techniques.

For a 2D generator, we chose StyleGAN-XL (Sauer et al., 2022) since it achieves state-of-the-art visual quality on ImageNet, as measured by FID (Heusel et al., 2017). We took the original checkpoint for $256^2$ generation from the official repository.[5] First, we generated 50,000 images with the model and computed their FID: this gave the value of 2.51 vs 2.26, reported by the authors, which is a negligible difference.

After that, we used the original codebase of 3DPhoto (Shih et al., 2020) to synthesize the 3D variations of 10k random images. We kept all the hyperparameters the same. In 3DPhoto, there is no spherical camera parametrization, this is why we had to simulate it the following way. We assumed that the sphere center lies at the median depth value from the original camera position (which is $(0, 0, 0)$), and had been rotating the camera around that point. As discussed in §4, we used the narrow camera distribution to sample the points. Namely, we used the normal distributions with standard deviations of $\sigma_{\text{yaw}} = 0.3$ and $\sigma_{\text{pitch}} = 0.15$ for sampling rotation and elevation angles, respectively.

---

[5]https://github.com/autonomousvision/stylegan_xl

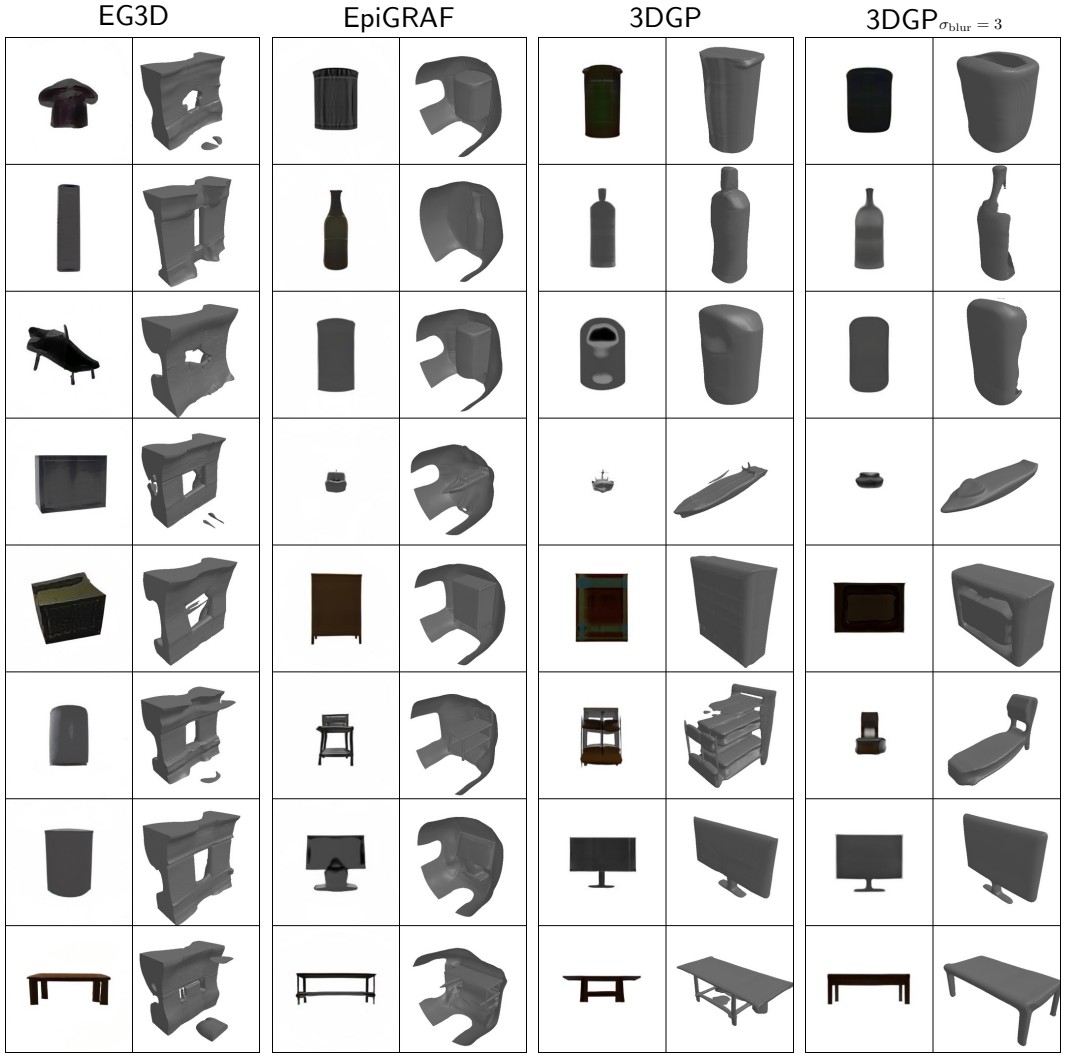

Figure 16: Random samples (seed 1, classes 1-8) for ShapeNet $256^2$ for our method and the baselines.

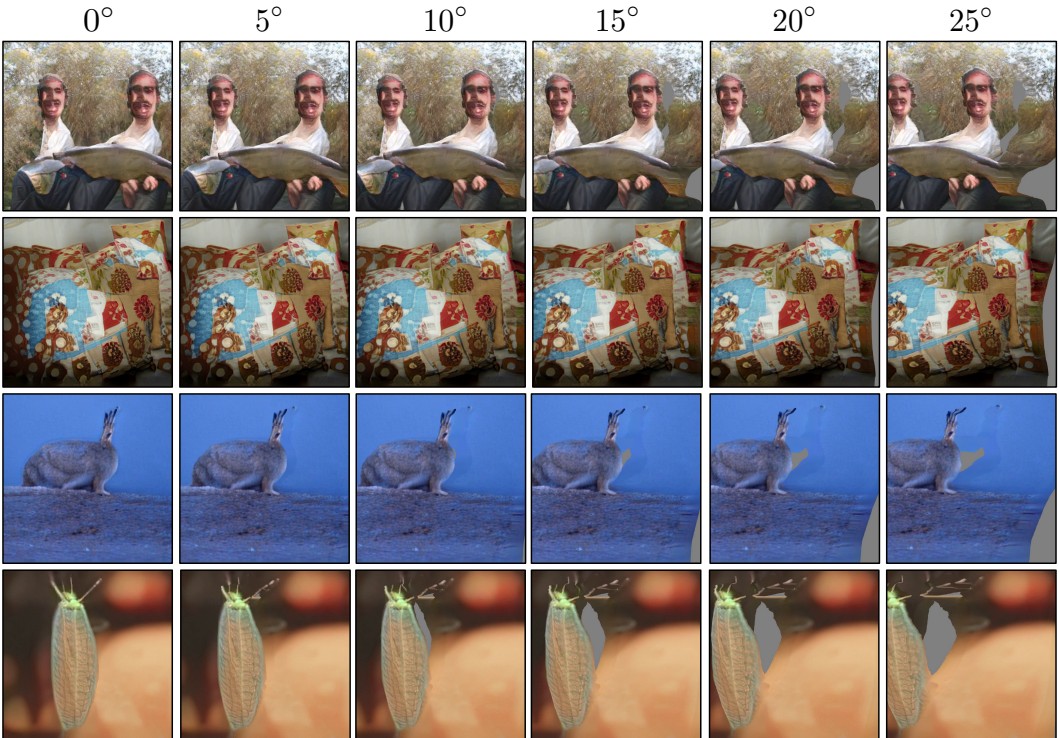

Figure 17: Random samples from StyleGAN-XL (Sauer et al., 2022) paired with 3D Photo Inpainting (Shih et al., 2020). For large camera movements, there appear severe artifacts, like gray areas and interpolation artifacts.

For negligible camera variations, this strategy produces excellent results. While it is not perfectly view consistent, it inherits state-of-the-art image quality from the 2D generator, which was used to synthesize the original images. But for larger camera variations, the quality becomes to deteriorate very quickly. Not only noticeable inpainting artifacts start to appear, but also outpainting problem emerges. The model was not trained to perform image extrapolation or inpaint large regions and fails to fill the holes which appear in larger camera movements. We provide the visualizations for it in Fig. 17.

## J    ENTROPY REGULARIZATION

After the submission, we explored another strategy to regularize the camera generator and observed that it is more flexible and easier to tune. Intuitively, the strategy is to maximize the entropy of each predicted camera parameter $\phi_i$:

$$\mathcal{L}_{\varphi_i} = H(\varphi_i) = \mathop{\mathbb{E}}_{p_G(\varphi_i)} \left[ -\log p_G \varphi_i \right], \tag{9}$$

where $p_G(\varphi_i)$ is the camera generator's distribution over $\varphi_i$, $H$ is differential entropy.

To do this, we used the POT package Flamary et al. (2021) to minimize the Earth Mover's Distance with the uniform distribution:

$$\mathcal{L}_{\varphi_i} = \text{EMD}(p_G(\varphi_i), U[m_{\varphi_i}, M_{\varphi_i}]), \tag{10}$$

where $\text{EMD}(P, Q)$ denotes the Earth Mover's Distance between the distributions $P$ and $Q$, $U[a, b]$ is the uniform distribution on $[a, b]$ and $m_{\varphi_i}, M_{\varphi_i}$ are minimum and maximum values for a given camera parameter $\varphi_i$.

In practice, we used 64 samples to approximate the EMD and a non-regularized Wasserstein distance.

