# OpenReview forum: "3D generation on ImageNet"
_ICLR.cc/2023/Conference — ICLR 2023 notable top 5%_

### Official Review · Reviewer_K59a · 2022-10-24

**Confidence:** 4
**Correctness:** 4
**Technical Novelty And Significance:** 3
**Empirical Novelty And Significance:** 3
**Recommendation:** 6

**Clarity, Quality, Novelty And Reproducibility:**

The paper is written clearly. Although there is no code provided, the results appear to be reproducible. The novelty of the paper is discussed in the section above.

**Strength And Weaknesses:**

### Strengths
* This paper is an empirical work, and the empirical results are relatively strong.
* There is no reliable quantitative measure of generated geometry, so the paper introduces Non-Flatness Score. NFS is a very simple heuristic, but it does seem to provide at least some insight into the flatness of generated geometry.
* The NFS values (Table 2) show that for unaligned datasets, previous work (e.g. EG3D) does not learn reasonable depth values, whereas 3DGP does learn reasonable depth values.
* The proposed camera regularization strategy is intuitive. Figure 7 is an illustrative figure in this regard.
* The inclusion of Appendix D (Failed Experiments) is very nice to see and much appreciated. The field would benefit substantially if sections like these were included in all papers.

### Weaknesses
* The novelty of this paper may or may not be viewed as a weakness. On the one hand, there are no groundbreaking ideas; the paper uses components that have been used before and combines them to obtain strong results. On the other hand, the exact method has never been used before; nobody else has combined these components in the precise manner that is being proposed here.
* For the ImageNet experiments, the method uses more time/compute than competing methods (28.2 days vs 18.7 days or 15.9 days; Table 2). Training for a similar amount of time as prior work would allow for a fairer direct comparison.
* When discussing GANs with external knowledge (at the top of page 4), the paper claims that “A similar technique is not suitable in our case as pre-training a generic RGB-D network on a large scale RGB-D dataset is problematic due to the lack of data.” However, it should be possible to train a generic (2D) RGB-D network on existing 2D image datasets by converting the 2D RGB images into RGBD images using monocular depth estimation; the paper effectively uses the same.

#### Minor Weaknesses
* The method appears to be quite sensitive to hyperparameters (e.g. those on page 7). This is somewhat expected, as other GAN-based methods also tend to be quite hyperparameter-sensitive.

### Questions:
* Existing 3D monocular depth estimation models are far from perfect. Are the failure cases from the monocular depth estimation network learned and reproduced by the generative network? This is mentioned briefly on Page 6 as motivation for the depth adaptor; it would be interesting to provide some more detail on this point.
* Do symmetries (e.g. those with respect to the position of the camera on the sphere) cause any issues with learning the camera distribution?
* With regard to the camera distribution, it seems strange to have the same camera distance for diverse ImageNet categories such as a goldfinch and a church. One can of course partially account for this by predicting different focal lengths, but have you tried learning a more flexible distribution of camera such as the inside of a sphere or a 3D annulus? Additionally, how does the distribution of learned focal lengths for different ImageNet classes look like?
* Given that the model learns a triplane representation, it should be possible to rotate all the way around the scene. However, all of the examples shown are front-facing views with small-to-medium camera movements. Does the model learn reasonable backsides for objects? For example, the backside of the horses shown at the bottom of Figure 5. It is not critical for the paper to reconstruct the backside of objects.



**Summary Of The Paper:**

This paper aims to learn a generative model of 3D scenes from large unaligned datasets. The paper argues that previous work only produced good results for well-aligned datasets (e.g. human and animal faces), and that directly applying such methods to unaligned data did not work well. In order to address this problem, the idea is to incorporate an off-the-shelf monocular depth estimator (in this case, LeReS) into their model architecture. In their proposed model, a discriminator aims to discriminate between an true RGBD image (the result of applying LeReS to a real unaligned image) and an RGBD image rendered from their triplane image representation. The discriminator also outputs an embedding vector which aims to represent the semantic content of the image, effectively distilling a feature model (in this case, a ResNet-50) into the discriminator. Additionally, the paper uses a more flexible camera parameterization than previous work: the model predicts a field-of-view and look-at position in addition to the position of the camera on a fixed-radius sphere.

The paper performs experiments on multiple datasets including a variety of LSUN categories and the full ImageNet dataset. It shows strong results compared to previous work using GANs (e.g. EG3D, EpiGRAF, etc.).


**Summary Of The Review:**

This paper proposes a method for learning 3D generative models from large datasets of unaligned images. The paper tackles an important
topic in unsupervised 3D reconstruction. Although the paper does not have any groundbreaking ideas, it combines ideas and models from the existing literature in a different manner from previous work. It tackles an important and often-overlooked problem (that of generalizing existing methods to real-world datasets) and produces relatively compelling results.

---

> ### Author Response · Authors · 2022-11-11
> **Responding to concerns and questions [Part 2/2]**
>
> Answers to the questions:
> - **About the generator inheriting depth estimator artifacts.** That is a nice question, and the depth adaptor is designed entirely to mitigate this issue. Our model has quite some geometry artifacts, and it is difficult to attribute them to either the depth estimator failure cases or the generator's synthesis problems. With the next submission update, we'll include samples showing the depth estimator's artifacts and our generator's geometry failure cases. We thank you for this remark.
> - **About the symmetries hampering the camera learning.** That is a difficult question, and we believe they are not. The thing is, the generator performs some implicit alignment of the underlying objects to fit them better — for example, you can observe from the videos, that 7/9 of the dogs look approximately into the camera (the samples there are random). At the same time, for real data this happens in only ~30% of the cases (see random images [here](https://www.dropbox.com/s/v8yhsdco0tccb8j/dogs-random.png?dl=0)). This way, it tries not to rotate the objects but only to rotate the camera. We suspect that happens because it is difficult for $\mathsf{G}$ to do rotations when it is parametrized as tri-planes produced by 2D convolutions.
> - **About focal length distribution.** We use tri-planes parametrization for our generator, which creates problems when using a large focal length or small radius: tri-planes define just a $[-0.5,0.5]^3$ volume cube of $512^3$ resolution. This way, zoomed-in close-up views would inevitably utilize fewer tri-planes features when rendered, resulting in low-resolution artifacts. Intuitively, this could be understood in the limit: imagine if we render with a camera radius of 0.0001, then we would interpolate features from just a $1 \times 1 \times 1$ part of the tri-planes volume. In this case, all the modeling load would be put on the tri-planes MLP, which is too small (a single layer of 64 neurons) to perform any meaningful synthesis on its own. This is why the focal length distribution for different ImageNet classes looks very similar: the camera generator learns the distribution with as large FoV as possible rather than doing fine-grained FoV control. We included this discussion with the focal length distribution plot in Appendix A (i.e., the "Limitations" section).
> - **About 360-degree generation.** We parametrized our camera position distribution to always be on the frontal hemisphere, that is why we never render the back side of any objects. We did this because in our datasets, there are no backsides of the objects (or they appear with too little probability). This is why the generator has no chance to learn the backsides of the objects.
>
> Thank you once again for your review and for your questions, which we find to be very in-depth. Please let us know if some of the above would benefit from additional clarifications or if some new questions or concerns have arisen on your side. We would be happy to discuss this. Also, let us know if there is anything within our hands we can do that you think could improve our submission. Also note that we've improved the writing in several parts of the work and added the reproducibility/ethics statements. And we will also release the source code and the checkpoints.

---

> > ### Comment · Reviewer_K59a · 2022-11-14
> > **Thank you for the clarifications**
> >
> > Thank you for taking the time to respond comprehensively to my questions and those of the other reviewers.
> >
> > The responses answered my questions quite well.
> > * The method is clearly more compute-efficient than expected. The attached FID convergence plots make this clear.
> > * The responses about symmetries and focal lengths addressed my concerns on those points.
> > * Regarding an RGBD GAN, thank you for the clarification. I now agree that it would be more complex to train this GAN than the approach proposed in this paper.
> > * It makes sense that the dataset and technique is not designed for 360-degree generation as the training data does not support learning backside views.
> > * Thank you for planning to release the source code and checkpoints. These will be very helpful to the community.
> > * There have been meaningful updates to the writing and the inclusion of ethics and reproducibility statements.
> >
> > The other reviewers have also raised interesting points.
> > * A comparison with 3D Photo Inpainting (reviewer yhJP) would be very insightful.
> > * The connection of the flatness issues to the tri-plane generator (reviewer pJ3E).
> >
> > Overall, I (and to my understanding the other reviewers) have not found any major flaws in this work and I maintain in favor of acceptance with a score of 6.

---

> > > ### Author Response · Authors · 2022-12-08
> > > **An update notice**
> > >
> > > Thank you once again for your review and the response. At the end of the previous discussion stage, we had updated our draft (summarized in [that message](https://openreview.net/forum?id=U2WjB9xxZ9q&noteId=3fJA7v6XHl)), and in this message, wanted to make sure that the update information has been brought to your attention. We’ve incorporated quite some experiments, evaluations and results, including the comparison with 3D Photo Inpainting and non-tri-plane-based 3D generators.

---

> ### Author Response · Authors · 2022-11-11
> **Responding to concerns and questions [Part 1/2]**
>
> We are grateful for your in-depth review of our work and we appreciate the time which you spent studying it.
>
> Responses to the raised concerns:
> - **About the novelty.** Well, "novelty" feels like a subjective and very tricky substance to use to measure a paper's merits. Our work makes a decent step towards solving a significant unsolved problem that the community greatly cares about (3D generation from in-the-wild data). As for the sheer novelty, several components have never been explored (to the best of our knowledge) in prior works: 1) depth adaptor (or even any adaptor?) on top of the generator to handle the distribution gap (one can loosely relate to differentiable augmentations here); 2) the exact form of the camera generator (as shown in Fig. 9); 3) the exact form of its regularization; 4) the idea to incorporate classifier's external knowledge into D via distillation. It took quite some thinking and experimentation to arrive at each of these components, but what is more important — each of them makes us one step closer to solving the problem. We believe that future works on this topic will find good value in our study, which is the main thing that matters. The most popular works in the field are very often a combination of existing ideas (e.g., even ResNet or Transformer), and it is the value they bring which makes them impactful.
> - **About the compute time.** Well, in fact, our scores would be even better if we'd train for the same amount of time as EG3D. Our model stopped improving right after 17.5 A100-days (but after the submission, we found the issue with it and now have ~25% better FID of 20.5), and our FID at 17.5 was 25.63 vs 25.6 of EG3D at 18.7 A100-days. We reported our compute time as 27.5 A100-days instead of 17.5 A100-days, since we used the checkpoint from after 27.5 A100-days because its samples felt somehow better in terms of geometry. But if someone tries to reproduce the model, they can safely stop after 17.5 A100-days with even a slightly better score. We provide the FID convergence plots via [this link](https://www.dropbox.com/s/p19wiogx040gpsw/fid-convergence.png?dl=0).
> - **On building an RGB-D classifier to distill from.** While it is possible in theory, it is way too complicated in practice. There are three issues with that: 1) one would need to spend a lot of resources to train such a classifier (e.g., modern ImageNet classifiers, especially ViT-based, are even more expensive to train than our 3D generators); 2) one would be limited in terms of which classifier it has access to (since one needs to re-train each time, while for us it is just 2 lines: "import timm; model = timm.create_model(...)"); and 3) one would be unable to distill from huge-scale models, like CLIP (which was giving us ~10% FID in the preliminary experiments, but we didn't use it since it was just too much external data and hence unfair comparison). Currently, our distillation strategy is just ~10 lines of architecture/losses code + ~70 lines of preprocessing, but training a classifier is much more challenging. Also, what if the discriminator is trained patch-wise (like in our case)? Should we train the classifier also patch-wise? Should we use the same patch annealing strategy? Should we use the same set of augmentations for both the classifier and the discriminator (if it will use differentiable augmentations)? What if we want to distill from Mask R-CNN features instead? We'll have to re-train it on RGB-D, which would be a headache. In this way, our form of knowledge distillation is way more discreet and easy to incorporate into existing pipelines.
> - **Hyperparameters sensitivity.** That is true, but the main reason (as you mentioned) is that we build on top of the components (GANs and NeRFs), which are very sensitive to hyperparameters themselves.
> But also note that we use the same hyperparameters for all the datasets.

---

### Official Review · Reviewer_GoQR · 2022-10-24

**Confidence:** 4
**Correctness:** 4
**Technical Novelty And Significance:** 3
**Empirical Novelty And Significance:** 3
**Recommendation:** 3

**Clarity, Quality, Novelty And Reproducibility:**

- 3DGP can be a confusing name given EG3D is a previous work. The authors might want to reconsider this.

- The pdf I receive has some lines on page 10 that violate the page format.



**Strength And Weaknesses:**

** Strength **

- The use of depth priors is an interesting idea to guide 3D-aware generative model.


** Weakness **

1) The use of the off-the-shelf depth estimator could, however, result in unfair comparisons as the depth estimator relies on another datasets for training. To be fair, the depth estimator has to be trained on the same dataset for the generative model, but this might not be possible on ImageNet.
The effect of these additional data should be properly discussed.

    a) Geometry prediction of the original depth estimator should be provided.

    b) Compare a) to the geometry generated by the proposed method.

    c) Compare the geometry generated by different depth estimators, and depth estimators trained on different datasets. Again, this point is not convincing to me still because the use of the additional data.


2) More results on ImageNet should be presented as claimed in the paper title. For example, out of the 1000 classes on ImageNet, how many categories have the authors experimented?
To evaluate the geometry quality, the authors might want to refer to real object datasets such as ScanObjectNN, and compare the reconstructed geometry with these objects in point cloud format with FID evaluation.

    [a] Uy et al., Revisiting Point Cloud Classification: A New Benchmark Dataset and Classification Model on Real-World Data, ICCV 2019.


**Summary Of The Paper:**

The paper presents a new method that learns a 3D-aware generative model on unaligned image datasets like ImageNet.
The authors propose to use an off-the-shelf depth estimator to guide 3D generator. The estimated depth is concatenated to the real image and feed to discriminator to guide the generated image and depth from the generator. Additionally, a learnable camera model based on the ball-in-sphere geometry is proposed to support the diverse camera settings in unaligned data, which includes position, fov, and lookat parameters.
The results show that the method can generate realistic images with depth on ImageNet dataset.

**Summary Of The Review:**

The paper itself is an interesting read and provides a step toward making 3D-aware generative models more practical and robust. My concern is that the experiments are unfair and could lead to confusion. I am happy to raise my score if the authors could convince me about the actual performance of the model given the off-the-shelf depth estimator.

---

> ### Author Response · Authors · 2022-11-07
> **Discussing the evaluation pipeline to address the raised concerns**
>
> We are grateful to you for studying our work and for the review. We believe that your concerns are reasonable and that we can resolve them — but for this, we would like to first discuss the evaluation/argumentation strategy with you to make sure that it aligns with your vision.
>
> Currently, we have the following plan to address your concerns.
>
> **1. Understanding the importance of the additional data used to train the depth estimator.**
>
> The LeReS depth estimator (which we used) was trained on the combination of 4 datasets (where only a part of each dataset was used):
> - 18.2k images from HRWSI — outdoor city imagery (e.g., buildings, monuments, landscapes).
> - 95k frames from 211 videos from DiverseDepth — clips from in-the-wild movies and videos.
> - 42k images from Holopix50k — diverse, in-the-wild web images. It is the most similar to ImageNet.
> - 135k views of 534 scenes from Taskonomy — indoor scenes (bedrooms, stores, etc.).
> Here are 100 random images from each dataset source: [https://www.dropbox.com/sh/9tnl95s4p1h0ozi/AAAHgRqQJE40IDIB3tAjathqa?dl=0](https://www.dropbox.com/sh/9tnl95s4p1h0ozi/AAAHgRqQJE40IDIB3tAjathqa?dl=0).
>
> In our work, we trained each model (3DGP, EG3D, and EpiGRAF) on all 1000 categories from ImageNet. One can show that the depth estimator’s training data does not “help” our generator the following way. ImageNet mainly consists of animal images (482 classes out of 1000), but the above datasets mostly contain indoor/outdoor scenery and human photos. Then, their images are of no help to generate animals, which can be rigorously shown the following way:
>
> - Measure FID between the depth estimator’s training data and the ImageNet animal subset. If it is high — e.g., above 50 — then its training data is of no help to generate ImageNet animals (or could even be harmful, putting other generators in a more advantageous position).
> - Detect animal classes in the LeReS training data with MS-COCO pretrained Mask R-CNN (it can detect 10 most popular animal categories: dogs, cats, horse, elephant, etc.) to see how many images contain animals in it.
> - Compute FID scores on the animal subset for EG3D, EpiGRAF, our generator, and StyleGAN-XL. If the FID trend remains the same as in Table 2 of our paper, then our generator has learned to generate animals on its own, without the help of the depth estimator’s training data — since there are no animals in it (we expect to have less than 2%).
> - Provide visualizations for generated depth maps by 1) our model; 2) EG3D; 3) EpiGRAF, and 4) depth estimator — on 1) animal classes; 2) non-animal classes.
>
> **2. Geometry evaluation.**
>
> For this point, we plan to follow your suggestion and perform the following evaluation:
> - Train EG3D, our generator, and StyleGAN2 (as a 2D generator upper bound) on ShapeNet or Redwood-3DSCAN or ScanObjectNN or a similar dataset (after some exploration, we found the problem with ScanObjectNN as it has fewer objects — just 2309 — and does not seem to provide canonically oriented point clouds or corresponding camera positions, making it impossible to render it properly automatically) as you suggested. We will train our generator in two variants: with the depth maps predicted by LeReS and with the ground-truth depth maps (the latter one will additionally show that our generator does not get help from the training data used to train LeReS). EG3D will be trained with narrow and wide cameras. All the generators would be trained for the same amount of time (~3 days on 4 A100 GPUs).
> - Evaluate FID and Frechet Pointcloud Distance (FPD) from the tree-GAN paper. Real point clouds will be taken from ground-truth depth maps, while fake point clouds will be obtained by rendering the depth maps from each generator.
> - Our model is expected to have considerably better geometry (at least 30% lower FPD) and similar image quality (at most 5% higher FID) than EG3D (or EpiGRAF — whichever is stronger). We do not aim to have “better-than-EG3D” image quality because EG3D uses a 2D upsampler which damages the geometry and view-consistency while improving the image quality — i.e., it is something between a 2D and 3D generator.
> - Also, provide samples/depth maps visualizations for EG3D/Our generator/LeReS and ground truth for qualitative assessment and sanity checks.
>
> *Do you believe that if we will execute the above two evaluation pipelines **and** its results will coincide with the outlined expectations, then, from your judgment, our work will be good enough to pass the acceptance threshold? (Of course, we would also incorporate these results into the paper).*
>
> As to the paper naming — thank you for the observation, we did not think about it in this regard and will explore other alternatives. We also fixed the unfortunate formatting error and updated the pdf (and also re-worked the writing/structure in several parts of the paper and included the Reproducibility and Ethics statements).

---

> ### Author Response · Authors · 2022-11-23
> **Experiments and evaluation results**
>
> We have updated our submission with the new experiments and evaluations, which we outlined in our previous message and that are targeted to resolve your concerns:
>
> 1. *About the depth estimator's training data influence on the generator.* We explored this concern by investigating the training data of the depth estimator. We found that its training data contains very few animal images (in terms of sheer number of objects, estimated with Mask R-CNN and also via computing FID with ImageNet animals classes), while the training data of our generator is dominated by animals (e.g., ImageNet is ~50% animals, other datasets are 100% animals). Then we checked the performance of our generator (and the baselines) on the animal subset of ImageNet and the non-animal subset. We observed exactly the same performance trend as for our main experiments, which implies that there is no "data leakage" from the depth estimator, which would unfairly improve the performance of our method. The details are in Appendix G.
> 2. *Geometry evaluation*. We used ShapeNet* to render single-view frontal generations of multiple categories (to be as close as possible to practical scenarios). Then, we trained several methods and 3DGP with true and corrupted depth maps (with different levels of corruption). After that, we extracted meshes using marching cubes (using the setup from EG3D) and extracted the pointclouds from real and fake meshes. We measured FID and Frechet Pointcloud Distance, and observed that 3DGP produces both better visual quality and *much* better geometry than the other methods (but starts deteriorating when the depth maps get worse). The details/visualizations are in Appendix H.
>
> From our perspective, those results address your concerns well, but if you believe that there could be any additional clarifications from our side which would help to resolve your concerns — please, let us know.
>
> *The ScanObjectNN dataset, which you mentioned, turned out to have two issues: 1) it is very small, and 2) the pointclouds are quite sparse.

---

### Official Review · Reviewer_pJ3E · 2022-10-25

**Confidence:** 4
**Correctness:** 4
**Technical Novelty And Significance:** 3
**Empirical Novelty And Significance:** 3
**Recommendation:** 8

**Clarity, Quality, Novelty And Reproducibility:**

**Clarity**

This paper is clear overall. My specific doubts and concerns are mentioned in above section. Appendix is useful and informative.

**Novelty**

There are mainly 3 introduced modules: 1) depth-regularized GAN, 2) learnable camera formulation, 3) a knowledge distillation loss. All of them are interesting and novel for 3D GAN framework.

**Quality**

- Despite being novel and promising, the introduced novel modules aren't very easy to use and couple of additional modules (eg., adaptor, camera prior, additional hyper-parameters) need to be leveraged.
- Also, the quality of the generated 2D images far way behind the state-of-the-art 2D generators. (as shown in
 https://u2wjb9xxz9q.github.io./) Even though being view-consistent, there exist many obvious artifacts. Nevertheless, this paper is still an exciting first step in this challenging problem setting.

**Reproducibility**
The paper provides fine implementation details in both main paper and the Appendix. However, it would be still quite challenging to reproduce without the source code since the propose framework is relatively complex.

**Details Of Ethics Concerns:**

I didn't find any ethics issues even though I didn't check research integrity issues (e.g., plagiarism, dual submission).

**Strength And Weaknesses:**

**Strength**
- The biggest limitation of existing 3D GAN is the applicable data. This paper extends the 3D GAN to a more challenging setting where large-scale un-aligned dataset is used.
- The introduced components are very interesting and works well.
   1. The learnable camera formulation is very useful and can be a standard thing for follow-up works in this field. I think this is very important for models object-centric images in the wild.
   2. The motivation of using depth as regularization makes lots of sense. Even though it's not easy to leverage estimated depth data, the proposed method managed to being beneficial.
   3. The knowledge distillation loss is particularly useful for ImageNet.

**Weakness**

- My biggest concern is the visual quality. From the provided annoymous link, the generated images suffer from many artifacts. Even though the "flat" issue is addressed compared to EG3D, the visual quality is merely improved. Both of them fall far behind the 2D generators.
- Even though using depth is resonable, it also makes the framework more complicated. It's quite tricky to do the Adversarial Depth Supervision (ADS) and set the p(d) correct.
- More 3D GAN baselines besides EG3D (tri-plane) could be tested. It's possible that the "flat" issues are caused by the tri-plane representation.  Therefore, a volume-based or MPI-based 3D GAN could have very different results.
- The loss in Eq.1 is somewhat weird. To me, this loss is minimizing and maximizing the same thing simultaneously. Why does this loss make sense? What does it converge to? What's the optimal value for this task?

**Doubts**

- Is “Ball-in-Sphere” formulation good enough for any images in-the-wild? What stops us from making it more flexible? For example, why do we need the camera to stay on the hemi-sphere? Why do the loss weights for the three components (pos, fov, lookat) differ so big?
- How are the camera priors decided? In Fig.7 most priors are relatively uniform, but pitch/look-at are more like gaussian. It's unclear to me how to set these priors. Do we need to choose different priors when dataset changed?  Also, it's also unclear to me why residue based method fails so bad.
- Why squared loss for distillation? This is not the standard option for knowledge distillation. Does it outperform standard setting, ie., KL-divergence?
- Why not applying a depth loss on adaptor output? For example, applying LeReS on the rendered image to get the ground-truth depth and use it to supervise the output of adaptor.

**Writing**

- For 2D methods (StyleGAN2 (with KD)) in Tab.1-2, it's better to change their NFS score from 0 to N/A.
- The qualitative results from the link could be added to the paper (appendix is fine).
- The paper should be self-contained and thus R1 gradient penalty should be given in Appendix.

**Summary Of The Paper:**

This paper extends the 3D GAN to more diverse and un-aligned datasets (eg., ImageNet). This problem setting change is significant and very challenging, as we have learned from 2D fields (StyleGAN3--> StyleGAN-XL). This paper introduces several new modules to adapt an existing model EpiGRAF to this new setting including 1) depth-regularized GAN, 2) learnable camera formulation, 3) a knowledge distillation loss. Results on various dataset including ImageNet are promising in both rendering and geometry quality.

**Summary Of The Review:**

This paper tries to solve a very challenging task and proposes a legitimate framework consisting several new components. The results are promising and encouraging. I'm supportive overall and my current rating is 6 but not higher because the visual quality of the rendered images are not very satisfying. Also, the introduced framework is a bit complex since it incorporates several sub-modules/networks.

For the authors: I'm sorry and please point out if I overlooked anything. Happy to chat more from here.

---

> ### Author Response · Authors · 2022-11-09
> **Clarifying the concerns and discussing the experiments plan [Part 2/2]**
>
> **About our camera model**.
> - Our "Ball-in-Sphere" camera model is designed explicitly for object-centric datasets, assuming that the camera always looks "somewhere in the center". For the datasets with many outward-looking images (e.g., landscape panoramas), one would need a different parametrization with less control over the position and more control over the look-at position.
> - The only limitation which stops us from making the camera model more flexible is that it would be too difficult to optimize. NeRFs (especially in the tri-plane representation, which is bound to the $[-0.5,0.5]^3$ box) can represent only a limited part of the volume due to the capacity limits. When the camera is entirely unconstrained, then at initialization, it might appear in random space positions with random rotations and orientations. That is why we leave the camera position on the sphere and restrict the radius of the inner sphere to make the viewing frustum be positioned in approximately the same volume of the 3D space.
> - We set very small regularization coefficients to FoV and look-at because it will not help the camera generator $\mathsf{C}$ produce flat geometry even if it collapses its values to delta distribution. And interestingly, $\mathsf{C}$ chooses to control them even when it is not forced to do this via regularization. We set the regularization coefficient for FoV to be slightly larger to nudge $\mathsf{C}$ towards exploring the FoV distribution.
>
> **About the camera priors.**
> - We decided the camera priors to be as general as possible without letting the viewing frustum come out of the tri-planes $[-0.5, 0.5]^3$ box. And *use the same priors for all the experiments*. Our priors are specified in Figure 7: we selected them to be as general as possible: yaw (rotation) is uniform in $[-\pi/2, \pi/2]$, pitch (elevation) is spherical uniform in $[\pi/4, 3\pi/4]$, FoV is uniform on $[9, 21]$ (we had to restrict the upper bound since otherwise the viewing frustum will go out of the $[-0.5, 0.5]^3$ tri-planes box) and the look-at is uniform in the whole inner ball of radius 0.2 (we had to select the upper bound for the same reason as for FoV).
> - The residual bias fails because it does not constrain the camera generator enough: $\mathsf{C}$ simply needs to learn modeling y(x) = -x on a restricted range to diverge into delta distribution. Note that StyleNeRf made a similar observation about the residual camera generator diverging (see Appendix A2 of their paper).
>
> **About the squared loss for distillation.** The primary motivation to use L2 is to be able to distill from other models (like CLIP) as well. In fact, CLIP was giving us ~10% better FID for 2D generation, but we didn't include it at the end since it trains on a lot of extra data. We will run the training with KL distillation and report the results (and we will also report distillation from CLIP).
>
> **About applying depth supervision to adaptor's output.**
> That is a good suggestion, and we had a lot of internal discussion on this strategy when working on the project. The main reason is that our generator is trained patch-wise, and LeReS relies a lot on context to predict accurate depth. Moreover, it is pretty heavy to run: for example, it took us ~1 day on 4 A100 GPUs to perform inference for 40k images for SDIP Dogs $256^2$.
>
> Thank you for your suggestions on the writing: we changed the NFS score for 2D generators from 0 to N/A and added the $\mathcal{R}_1$ regularization description to Appendix B. We will add the qualitative results from the website with the next update, together with additional experimental results. We also substantially improved the writing and exposition in several parts of the paper (changes are highlighted in blue). Also note that we will release the source code and the checkpoints.
>
> Please, let us know if you would like to see any clarifications for anything in the above or seem to disagree with any of the above points — we would be happy to discuss. We will try to conclude the additional experiments as soon as possible. And thank you once again for your review, we genuinely find it good and very thorough.

---

> > ### Comment · Reviewer_pJ3E · 2022-11-17
> > **Response #2**
> >
> > The clarification of the camera stuff is very useful. In fact I actually love this idea and do believe that could be crucial for 3D GANs/generative models in the wild. I'm looking forward to the new results regarding the distillation using CLIP and other losses.
> >
> > I saw the new statement regarding the computing resources needed. I understand the difficulty of running experiments or ablating design choices under such scale, so as a reviewer I shouldn't be too demanding. The work is among the 1st batch of efforts to generalize 3D-aware GANs to ImageNet, which is interesting but not as straightforward as it appears to be. The proposed modules work well. I have checked the additional contents not only in my pool but also for other reviewers. I hope the authors can add most of these new stuff to the paper and finish the experiments they promised here and there. I am supportive overall and increased my final rating.

---

> > > ### Author Response · Authors · 2022-12-08
> > > **An update notice**
> > >
> > > We are grateful for your response and support of our work. By the end of the previous discussion stage, we have added a major update to our draft (summarized in [that message](https://openreview.net/forum?id=U2WjB9xxZ9q&noteId=3fJA7v6XHl)), including other knowledge distillation losses, writing improvements, comparisons to other methods and several new results and evaluations. We realize nove that we better should’ve posted this notice immediately after updating the draft, and we apologize if that draft update has been left unnoticed to you.

---

> ### Author Response · Authors · 2022-11-09
> **Clarifying the concerns and discussing the experiments plan [Part 1/2]**
>
> We are thankful for your time and for your review — it helps us improve our work. Below, we will address your concerns and clarify the doubts:
>
> **Visual quality.** Yes, our 3D synthesis visual quality is worse than for 2D generators. But there are several reasons for it, which we believe justify this situation:
> - *We used less compute.* Compared to 2D generators (reported in Table 2), our models are lightweight and take at least 2 times less compute to train (due to our resource constraints). And the compute plays a massive role in getting SotA results for generative models.
> - *We do not use a 2D upsampler.* Compared to EG3D, we do not use a 2D upsampler and do "real" 3D generation, while EG3D does "something in-between" 2D and 3D synthesis. Compared to EpiGRAF (the very recent SotA NeRF generator without a 2D upsampler), our method performs ~2 times better. In this way, our ideas improve a lot for 3D generation in terms of visual quality.
> - *3D GANs are in their infancy.* NeRF-based generators are very new and also quite different from conventional 2D generators. There are many under-explored areas in their design and training, and even major design choices (e.g., the NeRF backbone itself — tri-planes or MLPs or voxels) are yet to be established in the community. Also, they have much more hyperparameters, which are trickier to tune. For example, after the submission, we improved FID/IS on ImageNet from 26.5/73.1 to 20.5/117.1, respectively (BigGAN has 8.7/142.3) by simply increasing the EMA half-life period of the generator by 15 times (but there are still a lot of artifacts though…). A lot of things are yet to be explored for 3D generators.
> - *The task itself is challenging.* The gap between 2D and 3D generation on ImageNet turned out to be naturally just too large. It isn't easy to close it in a single project, especially when it is the very first work (to the best of our knowledge) on it.
> While we agree that a better image quality would make our work more valuable to the community — we also believe that the lower image quality does not make our ideas worse. It is the ideas and how they are explored that should ground the judgment.
>
> **About depth supervision being tricky to tune.** In fact, Table 1 shows somewhat the opposite: using p(d) in the [0.25, 0.5] range works equally well across 3 different datasets.
>
> **Testing more 3D GAN baselines.** That's a good suggestion, and we'll launch StyleNeRF (MLP-based 3D GAN) and VolumeGAN (voxel-based 3D GAN) on ImageNet and report the results for them.
>
> **About the gradient penalty.**
> - Our gradient penalty minimizes the function $\mathcal{L} = |g| + 1/|g|$, where $|g|$ is the input/output scalar derivative of the camera generator $\mathsf{C}$. The motivation is to prevent the collapse of $\mathsf{C}$ into delta distribution and the intuition is the following.
> - $\mathsf{C}$ can collapse into delta distribution in two ways: 1) by starting to produce the constant output for all the input values (this is being prevented by the first term $|g|$ and 2) by starting producing $\pm\infty$ for all the inputs, which are at the end converted to constants since we apply sigmoid normalization on top of its outputs to normalize them into a proper range (e.g., pitch is bounded in (0, $\pi$)) — this, in turn, is prevented by the second term $1/|g|$.
> - In fact, it took us quite some time, experimentation, and thinking to arrive at the current regularization form. Before that, we were experimenting with other forms of regularization, like maximizing the variance, or entropy, or pushing skewness/kurtosis to the one of the normal distribution. But the generator was constantly trying ways to "cheat" and collapse to delta distribution or a mixture of delta distributions (it very "likes" doing so since it can generate flat images and cheat the geometry in this regime).
> - The minimum value of the regularization term is 2, which is achieved when the gradient is constant and equals to 1, and we added its plot, together with the above discussion, in the Appendix.

---

### Official Review · Reviewer_yhJP · 2022-10-25

**Confidence:** 3
**Correctness:** 3
**Technical Novelty And Significance:** 2
**Empirical Novelty And Significance:** 2
**Recommendation:** 6

**Clarity, Quality, Novelty And Reproducibility:**

This paper explains its motivation and ideas well. The technical novelty is good. Some technical components in this work are of fair novelty. Many details are provided to ensure reproducibility. However, I found there are many hyperparameters (e.g., in Eq 5) that may make it hard to reproduce the results.

**Strength And Weaknesses:**

**Strength**
- The paper is well motivated as existing methods do not work well on non-aligned data.

- Using an off-the-shelf depth estimation as a generic prior is intuitive, and experiments show that such prior helps improve the generated quality.

- I find this paper easy and enjoyable to read and follow.

- The visual generation quality is significantly higher than compared methods.

- The learnable Ball-in-Sphere camera model does have a higher than a standard with 2 dof and works for objects that are not center-aligned.

- The experiments are carefully and extensively conducted. I appreciate that failed attempts are also mentioned in the appendix, which helps the audience and following researchers better understand this work.

** Weaknesses**

- The usage of an off-the-shelf depth estimator may make the comparison with other depth-estimator-free methods unfair. Plus, how do the performance and generalization of the monocular depth estimator affect the performance? Will the performance improve if trained with ground truth depth (for example if it is trained on a depth estimation dataset or a synthetic 3D dataset)?

- The gradient penalty is widely used and studied in GAN-based approaches, the authors simply extend that to the parameters of the camera model. How good is the method if multiple objects are in the scene? The authors imply that their model works well on "a scene
 consisting of multiple objects", but I cannot find any experiments verifying this claim.

- The so-called knowledge distillation for discriminator is more like an engineering trick and does not provide much technical insight to me. Moreover, the authors mention that "it can work with arbitrary architectures of the discriminator," but it can only work on a discriminator that produces features with the same dimension as ResNet50. I am also wondering if this L2 KD loss works better than the mostly used KL divergence loss.

- This paper should compare with a 3D photo baseline [a]. Combining a vanilla 2D image generator with such 3D photo generation methods could also produce realistic 3D image synthesis results while it seems to generalize on a larger range of images.  I wonder how the performance differs for these two different generation paradigms. Will the proposed method have a higher multi-view consistency? Will the proposed fail while the 3D photo methods work better in certain situations? Thus, this line of work should also be discussed.

[a] 3D Photography using Context-aware Layered Depth Inpainting. CVPR 2020.



**Summary Of The Paper:**

This paper introduces the first 3D generator that works on 3D datasets without alignment. To this end, conditioning on an off-the-shelf monocular depth estimation approach, the proposed method incorporates generic depth priors to facilitate 3D generation. Moreover, richer camera models are taken into account, ensuring more realistic generation results. Finally, with a distillation mechanism,  a pre-trained image classification network provides further supervision to stabilize and improve training. Experiments on standard datasets validate the effectiveness of the proposed method.


**Summary Of The Review:**

Based on the aforementioned pros and cons, I vote for a weak accept of this paper.

---

> ### Author Response · Authors · 2022-11-08
> **Discussing the experiments plan**
>
> In the following days, we plan to perform the following experiments/evaluation:
> - We will launch the experiments on ShapeNet (or a similar dataset) for EG3D, 3DGP and StyleGAN2. For 3DGP, we’ll use ground truth depths with different levels of corruption to see how it affects the performance. We believe that using GT depths should improve the geometry considerably (and one does not need the depth adaptor in such a case anymore). Still, the main goal of the work is what to do when only inaccurate depths are available since our target goal is large-scale in-the-wild datasets.
> - We will compare the 3D Photo Inpainting method on top of StyleGAN-XL (SotA ImageNet generator) in terms of FID and side FID views — we think that it will have slightly worse view consistency and considerably worse side views (due to its direct reliance on the predicted depth).
> - We will report the knowledge distillation training with the KL distance for ResNet50 instead of L2 and CLIP distillation with L2 for StyleGAN2 training.
>
> To improve the reproducibility, we will release the source code and checkpoints (we also updated the text to include the reproducibility statement and improved the writing in several parts of the work).
>
> Please, let us know if some of the above clarifications are not clear, or if you disagree with some of them or the proposed experiments/evaluation or expect them to be different. Also, let us know if you have any other concerns. We will be happy to discuss.

---

> ### Author Response · Authors · 2022-11-08
> **Clarifying the concerns**
>
> Thank you for your thoughtful review and for your suggestions. For some of your concerns, we would love to clarify them here below:
> - *About the comparison*. We would like to respectfully disagree with your concern about the comparison being unfair. We report the scores for other methods to give a reader the overall landscape of 3D generation on ImageNet and where 3DGP lands. We make it crystal clear that our approach relies on additional information and believe it would hurt the understanding if we remove the scores of other methods. Also, one could similarly argue that the comparison with, e.g. EG3D is unfair in the benefit of EG3D since it uses a 2D upsampler (i.e., hence having something between 2D and 3D generation) while 3DGP does not. To make things more clear, we added the disclaimer in Table 2 that our method uses additional information.
> - *About the camera model being more suitable for multiple objects*. Our camera parameterization lets the generator control where it looks on a 3D scene. And if we have a dataset containing, e.g. 50% images with 2 objects and 50% images with 1 object, then the generator would spend less capacity modeling this: it can always generate scenes with 2 objects and then just look into one of the objects when it wants to render an image with a single object. For a generator with the traditional camera parametrization, that wouldn't be possible since it is bound to always look into the center of the scene. On ImageNet, there is a Pomeranian dog class, where sometimes there appear images with 2 dogs on them (<5% of the cases — see 100 random images from it via [this link](https://www.dropbox.com/s/9gutzubr0nlypc4/imagenet-pomeranian.png?dl=0)). And our generator is able to learn this behavior and synthesize a 3D scene with 2 dogs on it (despite it being so rare): see [this link](https://www.dropbox.com/s/bdpe0y0w25vaqbq/pomeranian-generation.mp4?dl=0).
> - *About the gradient penalty*. Gradient penalties are indeed popular in GANs, but in a very different context — on top of the discriminator to stabilize training. In our case, we use it to prevent the camera generator from diverging into delta distribution. In fact, it took us quite some time, experimentation, and thinking to arrive at the current regularization form. Before that, we were experimenting with other forms of regularization, like maximizing the variance, or entropy or pushing skewness/kurtosis to the one of the normal distribution. But the generator was constantly trying ways to “cheat” and collapse to delta distribution (it likes doing so since, in this regime, it is able to generate completely flat images).
> - *About knowledge distillation being just an engineering trick*. We believe that this is the statement which we disagree with the most. It might look like a simple engineering trick after one saw it, but why didn’t it then become popular before when it is working considerably better and being more stable than the currently established “initialize D from a classifier and freeze the weights” strategy (see Table 4). It is also more general and much easier to implement (it is literally ~10 lines of code in terms of losses/architecture changes + ~70 lines of data pre-processing scripts), preserving the freedom over D architecture (which was important in our case since our model trains patch-wise). We believe that it is something that could receive a lot of adoption in the community. We didn’t use KL distance, and preferred the L2 loss to make the technique more general, so that one can experiment with non-classifier models as well (e.g., CLIP). We will compare with KL distance and also report the result for CLIP distillation (in our preliminary experiments, it was giving ~10% better results than ResNet50, but we couldn't keep it since it uses too much external data) for StyleGAN2 training. Also note that our technique is applicable on top of *any* discriminator and not just those which have the same dimensionality as ResNet50 — since we construct a separate 2-layer MLP head for the discriminator to predict the features (and in fact, our discriminator's feature dimensionalities do not match those of ResNet50).

---

> > ### Comment · Reviewer_yhJP · 2022-11-18
> > **Thank you for the response**
> >
> > I appreciate the detailed clarifications provided by the authors. Most of my concerns have been resolved, including the use of extra information, the generation of multiple objects, and novelty. I look forward to the experiments compared with 3D photo inpainting approaches. It would be interesting to have some in-depth analysis of the pros and cons of these two paradigms, and maybe some further research could take the best of both worlds.
> >
> > As for the technical contributions of gradient penalty and knowledge distillation, I encourage the authors to be clearer in the final version. For example, to answer the question, "why didn’t knowledge distillation then become popular before?" it is recommended to conduct more analysis (more than empirical studies) and look into the generalizability of it in other tasks.
> >
> > Otherwise, I do not have more concerns about the submission and will keep the acceptance rating.

---

> > > ### Author Response · Authors · 2022-12-08
> > > **An update notice**
> > >
> > > We wanted to thank you for your response, and just wanted to inform you that at the end of the previous discussion stage, our draft had been substantially updated, summarized in [that message](https://openreview.net/forum?id=U2WjB9xxZ9q&noteId=3fJA7v6XHl) (we now realize that we should’ve better post the notice immediately after updating it). We’ve added the comparisons to several new methods, including 3D Photo Inpainting: the problem we found is that it works well only for small camera movements, producing artifacts even for ~$10^\circ$ camera deviations. For knowledge distillation, exploring it for other tasks seems to be quite out of scope for our work — even in its current form, we had to move its key results on 2D generation into Appendix due to space limitations.

---

### Author Response · Authors · 2022-11-19
**Update summary**

We would like to thank all the reviewers for their valuable feedback, it helped us to substantially improve our submission. Below, we summarize our update (highlighted in blue in the draft).

1. We conducted a study on LeReS' pre-training data, its connections to ImageNet, and how it relates to our generator's performance. We showed that there is no implicit "data leakage" from LeReS' training data into our generator, which could potentially lead to unfair comparison. It is presented in Appendix G.
2. We created a new dataset from ShapeNet $256^2$ with single-view frontal camera distribution and evaluated the visual and geometry quality of EG3D, EpiGRAF, 3DGP and its variations in terms of FID, FPD (Frechet Pointcloud Distance), and shape visualizations. Our method considerably outperforms EG3D (which learns to invert the geometry) and EpiGRAF (which entangles the background). This study is presented in Appendix H.
3. We tested knowledge distillation with 1) KL distance on logits rather than $L_1$ on activations; and 2) CLIP instead of ResNet50. $L_1$ performs slightly better than KL, and CLIP performs slightly better than ResNet50. The results are in Fig 11.
4. We launched StyleNeRF and VolumeGAN for conditional ImageNet $256^2$ generation as they represent methods that are built on non-tri-plane-based NeRF backbones (MLPs and voxel grids, respectively). However, they produced quite low visual quality (${\approx}2$ times worse than 3DGP or EG3D in terms of FID). The results are in Tab. 2. and the website.
5. We ran 3D Photo Inpainting [1] on top of StyleGAN-XL generations and observed that it produces good results for small camera variations but leads to a lot of artifacts for even $10^\circ$-degrees camera changes. The results are in Tab. 2, the website, and Appendix I.
6. We added the visualizations of the samples, generated depths maps and depth maps predicted by LeReS on ImageNet in Appendix E
7. We included the clarifications and additional exposition arised from our discussion with the reviewers and also substantially reworked the writing. We also found that increasing the generator's EMA half-life parameter improves FID/IS scores for our model by ~20/40%, respectively.
8. We added the Reproducibility and Ethics statements. Also, note that we will release the source code and the checkpoints.

Thank you once again for your reviews, and please let us know if you have any other concerns or how we could improve our submission further.

[1] 3D Photography using Context-aware Layered Depth Inpainting

---

### Decision · Program_Chairs · 2023-01-20

**Decision:**

Accept: notable-top-5%

**Justification For Why Not Higher Score:**

N/A

**Justification For Why Not Lower Score:**

This is a very challenging problem that is of interest to a large subset of the ICLR community; a paper that makes headway on it merits extra attention. I waffled between "spotlight" vs. "oral," given the initially lower scores that this paper received, but in light of the thorough author response and the final opinions of the reviewers, I think it's reasonable to consider an Oral for this paper.

I marked "This decision can be bumped down" for the question below this one b/c I wouldn't mind if this paper were downgraded to a spotlight, should the SAC have a different opinion.

**Metareview: Summary, Strengths And Weaknesses:**

This paper presents techniques for learning a 3D generative model given only 2D training data; it goes beyond prior work in that this 2D data does not require foreground/background masks nor (more importantly) does it require the 2D training imagery to depict objects from the same viewpoint.

Strengths: reviewers agreed that the problem addressed by this paper is very important, and the paper makes a significant step beyond prior work by not requiring aligned training images. The proposed method achieves higher-quality visual results than prior work as well. In addition, reviewers believed that some of the technical components contributed by the paper could be useful in other contexts, as well: the learnable camera formulation and the knowledge distillation loss.

Weaknesses: some reviewers commented that the gradient penalty & knowledge distillation loss contributions could be viewed as minor engineering tricks (though this concern was well-addressed in the authors' rebuttal). Reviewers also identified some visual artifacts in the generated results and wanted to see how the method would fare when generating views of all sides of an object, as opposed to just the front (which should be possible with this method). There was also a request to compare against additional 3D GAN baselines beyond EG3D, and some concerns about how well the priors used would generalize to new datasets.

After reading the author responses, three reviewers commented that their concerns had been addressed and were in favor of acceptance.
One reviewer gave an unusually low score (3), given the length of review / magnitude of concerns raised.

**Note From Pc:**

if the above contains the word "oral" or "spotlight" please see: "oral" presentation means -> notable-top-5% and "spotlight" means -> notable-top-25%. As stated in our emails, we are disassociating presentation type from AC recommendations